# SAIGE-GENE+ improves the efficiency and accuracy of set-based rare variant association tests

Wei Zhou [1,2,3,10] ✉, Wenjian Bi [4,5,6,10] ✉, Zhangchen Zhao [5,6,10], Kushal K. Dey [7], Karthik A. Jagadeesh [7], Konrad J. Karczewski [1,2,3], Mark J. Daly [1,2,3,8], Benjamin M. Neale [1,2,3] and Seunggeun Lee [9] ✉

**Several biobanks, including UK Biobank (UKBB), are generating large-scale sequencing data. An existing method, SAIGE-GENE, performs well when testing variants with minor allele frequency (MAF) $\leq$ 1%, but inflation is observed in variance component set-based tests when restricting to variants with MAF $\leq$ 0.1% or 0.01%. Here, we propose SAIGE-GENE+ with greatly improved type I error control and computational efficiency to facilitate rare variant tests in large-scale data. We further show that incorporating multiple MAF cutoffs and functional annotations can improve power and thus uncover new gene–phenotype associations. In the analysis of UKBB whole exome sequencing data for 30 quantitative and 141 binary traits, SAIGE-GENE+ identified 551 gene–phenotype associations.**

UKBB recently released whole exome sequencing (WES) data[1], allowing study of rare variant associations for complex phenotypes. However, best practices remain unclear for rare variant tests in large-scale biobanks. A common practice is to test all rare (MAF $\leq$ 1%) loss-of-function (LoF) and missense variants, but this approach can lose power if associations are enriched in very rare variants or certain functional annotations. To improve power, researchers can restrict tests to rarer variants, such as variants with MAF $\leq$ 0.1% or MAF $\leq$ 0.01%. Another approach is to incorporate functional annotations. To incorporate multiple MAF cutoffs and functional annotations, multiple tests are needed for each gene or region, and results need to be combined using minimum $P$ value or Cauchy combination method[2,3].

Currently, SAIGE-GENE[4] is the only method developed to conduct variance component set-based tests, such as SKAT[5] and SKAT-O[6], for unbalanced case–control phenotypes in biobank-scale data. For example, in our evaluation, the most recent set-based test, STAAR[2], cannot control for type I error rates in the presence of case–control imbalance (Extended Data Fig. 1). Burden tests (such as implemented in REGENIE2 (ref. [7])) collapse multiple rare variants into a single variant, allowing the use of well-developed single-variant tests. However, Burden tests can have low power compared with SKAT and SKAT-O[6]. This was confirmed in our simulation studies (Extended Data Fig. 2). In analyses of UKBB WES data from 160,000 white British individuals (from the release

with 200,000 individuals), we found that SAIGE-GENE performed well when testing variants with MAF $\leq$ 1% (Fig. 1a), but inflation was observed in SKAT and SKAT-O in SAIGE-GENE when restricting to variants with MAF $\leq$ 0.1% or 0.01% if the case–control ratios were more unbalanced than 1:30 (Fig. 1a and Extended Data Fig. 3). Our type I error simulation studies (Supplementary Note; Methods) also showed the same inflation (Extended Data Fig. 4), suggesting that SKAT and SKAT-O in SAIGE-GENE can suffer from inflated type I error rates when restricted to variants with very low MAF.

In addition, computation cost is not low enough to test for multiple variant sets. For example, to test the largest gene (*TTN*) with 16,227 variants in the UKBB WES data with three maximum MAF cutoffs (1%, 0.1% and 0.01%) and three annotations (LoF only, LoF+missense and LoF+missense+synonymous), SAIGE-GENE required 164 CPU hours and 65 gigabytes (GB) of memory (Supplementary Table 1).

To address these issues, we propose SAIGE-GENE+. Although SAIGE-GENE uses various approaches to account for case–control imbalance, it cannot fully address the imbalance and sparsity in the data (Fig. 1a and Extended Data Fig. 4a). To reduce the data sparsity due to ultra-rare variants, before testing each variant set, SAIGE-GENE+ collapses variants with MAC $\leq$ 10 and then tests the collapsed variant together with all other variants with MAC > 10 (Extended Data Fig. 5; Methods). Collapsing has been commonly used for ultra-rare variants[8,9] by assuming those variants have the same direction of effects on phenotypes. We observed that the inflation is substantially reduced and all tests have well controlled type I errors in both simulated (Extended Data Fig. 4b) and UKBB WES analyses (Fig. 1b) for four exemplary phenotypes with case-control ratios from 1:32 to 1:267. The genomic control inflation factors also became closer to 1 (Extended Data Fig. 3).

Collapsing ultra-rare variants in SAIGE-GENE+ decreases the number of variants (Extended Data Fig. 6), leading to reduced computation time and memory usage (Fig. 2a, Supplementary Table 1 and Extended Data Fig. 7). To further reduce the computational cost, SAIGE-GENE+ extensively uses C++ with sparse matrix libraries, reads genotypes for all genetic markers in a set only once, and conducts multiple association tests corresponding to different MAF cutoffs and annotations (Supplementary Note).

[1]Analytic and Translational Genetics Unit, Massachusetts General Hospital, Boston, MA, USA. [2]Program in Medical and Population Genetics, Broad Institute of Harvard and MIT, Cambridge, MA, USA. [3]Stanley Center for Psychiatric Research, Broad Institute of Harvard and MIT, Cambridge, MA, USA. [4]Department of Medical Genetics, School of Basic Medical Sciences, Peking University, Beijing, China. [5]Center for Statistical Genetics, University of Michigan School of Public Health, Ann Arbor, MI, USA. [6]Department of Biostatistics, University of Michigan School of Public Health, Ann Arbor, MI, USA. [7]Department of Epidemiology, Harvard T. H. Chan School of Public Health, Boston, MA, USA. [8]Institute for Molecular Medicine Finland, Helsinki Institute of Life Sciences, University of Helsinki, Helsinki, Finland. [9]Graduate School of Data Science, Seoul National University, Seoul, Korea. [10]These authors contributed equally: Wei Zhou, Wenjian Bi, Zhangchen Zhao. ✉e-mail: wzhou@broadinstitute.org; wenjianb@pku.edu.cn; lee7801@snu.ac.kr

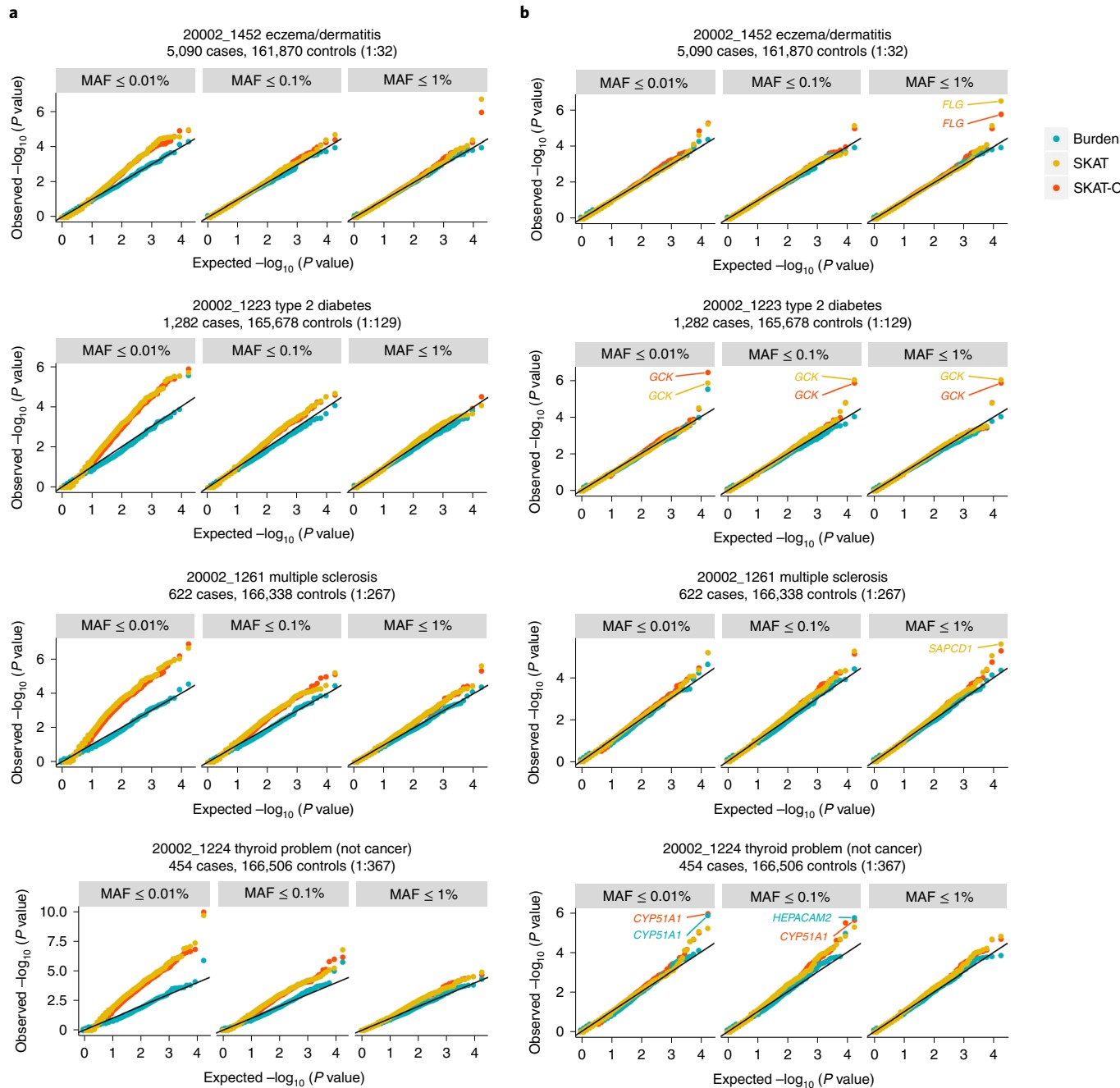

**Fig. 1 | Q–Q plots for Burden, SKAT and SKAT-O for four exemplary binary phenotypes in UKBB WES data using SAIGE-GENE and SAIGE-GENE+.**
**a**, SAIGE-GENE. **b**, SAIGE-GENE+. Burden, SKAT and SKAT-O tests were performed for 18,372 genes with missense and LoF variants with three different maximum MAF cutoffs (1%, 0.1% and 0.01%). Names of genes reaching the exome-wide significance threshold (two-sided $P < 2.5 \times 10^{-6}$) in SAIGE-GENE+ are annotated in the plots.

The computation time of SAIGE-GENE+ for performing all Burden, SKAT and SKAT-O tests was 1,407 times lower (9,851 min versus 7 min) and the memory usage dropped from 65 GB to 2.1 GB compared with SAIGE-GENE for testing association of the largest gene *TTN* (16,227 LoF+missense+synonymous variants) with basal metabolic rate (Supplementary Table 1). To perform SKAT-O tests for 18,372 genes in randomly selected 150,000 samples with three MAF cutoffs (1%, 0.1% and 0.01%) and three variant annotations (LoF only, LoF+missense and LoF+missense+synonymous), SAIGE-GENE+ required 78.6 CPU hours (18.8 CPU hours for fitting the null mixed model using a full genetic relationship matrix

(GRM) as Step 1 and 59.8 CPU hours for association tests as Step 2) and 4.8 GB memory (4.8 GB for Step 1 and 2 GB for Step 2) (Supplementary Tables 2 and 3 and Extended Data Fig. 8). In addition, when a sparse GRM instead of a full GRM was used in Step 1, the time and memory usage dropped dramatically (<1 min and 0.61 GB) (Supplementary Table 2, Supplementary Note, Extended Data Fig. 9 and Supplementary Figs. 1 and 2) and treating covariates as offset leads to a further decrease in the computation time (Supplementary Table 4). We also compared the computation cost of SAIGE-GENE+ and REGENIE2 (Supplementary Tables 2 and 3, Supplementary Note and Extended Data Fig. 8).

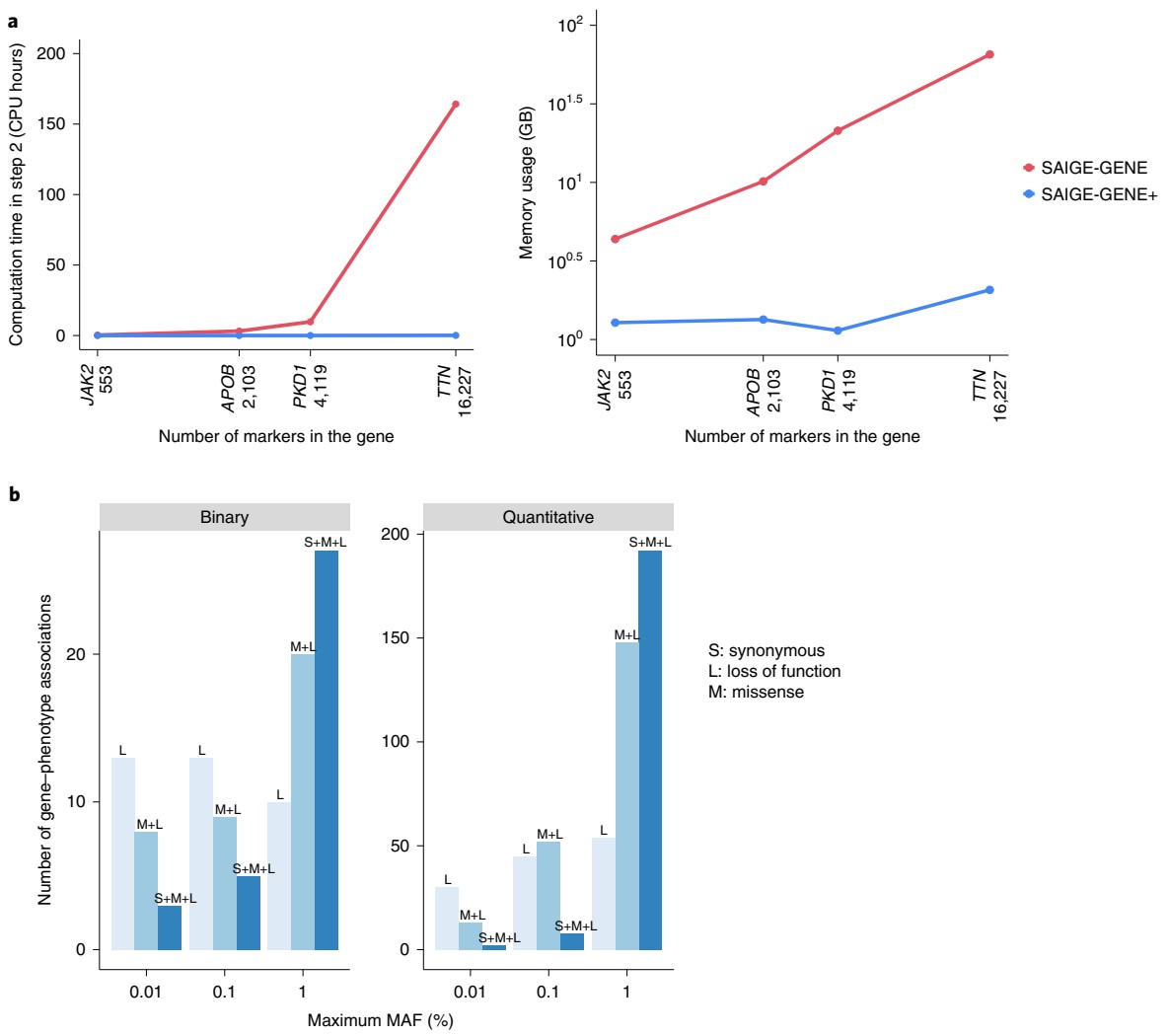

**Fig. 2 | Performance of SAIGE-GENE+ in UKBB WES data. a**, Computation time and memory of the gene-based tests (Step 2; Methods) in SAIGE-GENE and SAIGE-GENE+ for four genes with different numbers of variants. The SKAT-O tests were conducted with three maximum MAF cutoffs (1%, 0.1% and 0.01%) and three variant annotations (LoF only, LoF+missense and LoF+missense+synonymous) and combined using the Cauchy combination or minimum *P* value approach. Plots are in the $\log_{10}$ –$\log_{10}$ scale. Details of the numbers and genes are presented in Supplementary Table 1. **b**, Most significant variant sets across the three different MAF cutoffs (1%, 0.1% and 0.01%) and three functional annotations (LoF (L) only, LoF+missense (M+L) and LoF+missense+synonymous (S+M+L)). Distribution of variant sets with the smallest *P* values among 551 significant gene–phenotype associations identified by SAIGE-GENE+ in the analyses of 30 quantitative traits and 141 binary traits in the UKBB WES data.

By collapsing ultra-rare variants, SAIGE-GENE+ can have more significant *P* values than SAIGE-GENE. We applied both methods to 37 self-reported binary phenotypes in the UKBB WES data. We observed 27 significant gene–phenotype associations in which SKAT-O *P* values in SAIGE-GENE+ were more significant than SKAT-O *P* values in SAIGE-GENE (Supplementary Table 5). For example, *BRCA2* for breast cancer with MAF ≤ 0.1% had a *P* value of $7.62 \times 10^{-8}$ in SAIGE-GENE+ and $1.65 \times 10^{-3}$ in SAIGE-GENE, and *GCK* for diabetes with MAF ≤ 0.1% also had a more significant *P* value ($1.22 \times 10^{-13}$) in SAIGE-GENE+ than in SAIGE-GENE ($P = 4.06 \times 10^{-6}$). More detailed discussion is provided in the Supplementary Note.

We evaluated the power of SAIGE-GENE+ and SAIGE-GENE through simulation studies based on real genotypes of ten genes in the UKBB WES data (Supplementary Table 6, Supplementary Note and Methods). In all scenarios, SAIGE-GENE+ had higher or similar empirical power than SAIGE-GENE (Supplementary Table 7 and Supplementary Fig. 3) with increased median Chi-square statistics (Supplementary Table 8). In line with previous studies[6], our results showed that SKAT-O tests can have higher power than Burden tests

(Extended Data Fig. 2 and Supplementary Table 8). As expected, Burden test *P* values were highly concordant in SAIGE-GENE+ and REGENIE2 (Pearson's correlation $R^2 = 0.99$ for $-\log_{10}(P$ value)) (Supplementary Fig. 4). In addition, the simulation results suggested that incorporating multiple functional annotations (LoF, LoF+missense and LoF+missense+synonymous) and maximum MAF cutoffs (0.01%, 0.1% and 1%) can increase power compared with using only a single MAF cutoff (1%) on one set of function annotation (LoF+missense+synonymous) (Supplementary Fig. 5 and Supplementary Table 8).

We applied SAIGE-GENE+ to 18,372 genes in the UKBB WES data with 160,000 individuals of white British ancestry for 30 quantitative and 141 binary traits (Methods). We identified 465 gene–phenotype associations for 27 quantitative traits and 86 for 51 binary traits that were exome-wide significant with *P* values ≤ $2.5 \times 10^{-6}$ (Supplementary Tables 9 and 10), containing both known and potentially new associations (Supplementary Note). We created PheWeb-like server for visualizing these results (see Data availability)[10].

The UKBB WES data analysis showed that using lower MAF cutoffs can identify new associations in which the associations are highly enriched in rarer variants. For example, the association between *PDCD1LG2*, which encodes Programmed Cell Death 1 Ligand, and chronic lymphocytic leukemia became significant in tests restricted to variants with MAF $\leq 0.01\%$ and 0.1% ($P = 7.5 \times 10^{-7}$) compared with tests with all variants with MAF $\leq 1\%$ ($P = 5.4 \times 10^{-4}$) (Supplementary Table 11). The underlying reason could be that associations are enriched in the rarer variants, for example, the most significant variant has a MAF $3.4 \times 10^{-4}$ (rs7854303) (see the PheWeb-like visual browser). Using a MAF cutoff $\leq 1\%$ includes many noncausal variants, and thus decreases power. Moreover, including lower MAF cutoffs helped to replicate known associations such as *MLH1* for colorectal cancer and *CDKN2A* for melanoma (Supplementary Table 11). Due to multiple comparison burden, including lower MAF cutoffs can make marginally significant associations insignificant. For 141 binary phenotypes, 17 out of 92 (18.4%) associations were further identified with lower MAF cutoffs, while 9 (9.8%) became insignificant (Supplementary Fig. 6a and Supplementary Table 11). For 30 quantitative traits, 28 out of 465 (6%) associations were additionally identified, while 53 (11.4%) became insignificant (Supplementary Fig. 6a and Supplementary Table 12), suggesting that restricting association tests to rarer variants has a gain for binary phenotypes. In functional annotation categories, 184 associations were identified by testing LoF only; including LoF+missense sets identified 299 additional associations and including LoF+missense+synonymous sets identified 91 more associations (Supplementary Fig. 6b). These results are consistent with our simulation studies showing that empirical power increased when incorporating multiple functional annotations (Supplementary Fig. 5). We also investigated which variant set among the 551 significant gene–phenotype associations had the smallest *P* value (Fig. 2b). Interestingly, in variants sets with MAF $\leq 0.01\%$, those with LoF only generally had the smallest *P* values, while with MAF $\leq 1\%$, LoF+missense+synonymous sets tend to have the smallest *P* values.

In summary, our results demonstrate that incorporating multiple MAF cutoffs and functional annotations in exome-wide set-based association tests can help identify new gene–phenotype associations, and that SAIGE-GENE+ can facilitate this.

## Online content

Any methods, additional references, Nature Research reporting summaries, source data, extended data, supplementary information,

acknowledgements, peer review information; details of author contributions and competing interests; and statements of data and code availability are available at https://doi.org/10.1038/s41588-022-01178-w.

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

## Methods

**Collapsing ultra-rare variants.** Ultra-rare variants with MAC ≤ 10 are collapsed to a single marker, as illustrated in Extended Data Fig. 5. Like SAIGE-GENE, SAIGE-GENE+ allows incorporating weights for dosages or hardcalled genotypes of each marker. By default, to upweight rare variants, SAIGE-GENE+ calculates the weight for each variant using its MAF from a beta distribution beta($MAF$,1,25). SAIGE-GENE+ also allows users to specify per marker weights. The weighting scheme when collapsing ultra-rare variants is slightly different between these two. If the default MAF-based beta-weight is used, SAIGE-GENE+ first obtains the collapsed variant and assigns the weight based on collapsed variant frequency (Extended Data Fig. 5a). In particular, the dosage or genotype for each sample of the collapsed variant is assigned as the maximum raw dosage or genotype value among all ultra-rare variants carried by the sample. Then the weights of the collapsed variant and other less rare variants (MAC > 10) are calculated based on their MAF. Alternatively, if the per marker weights are provided by users (Extended Data Fig. 5b), the dosages or genotypes of the ultra-rare variants are first multiplied by the provided weights and then collapsed to a new variant whose dosage or genotype for each sample is assigned as the maximum values among the weighted dosages or genotypes of all ultra-rare variants carried by the sample. SAIGE-GENE+ also allows not incorporating any weights to set-based tests and collapses ultra-rare variants following the second scheme described above, as this is a special case that has equal weights for all variants.

**Aggregating multiple tests.** For each gene or region, $P$ values of multiple testing set corresponding to different maximum MAF cutoffs and functional annotations were aggregated using the Cauchy combination[2,3]. Note that the Cauchy combination does not work when a $P$ value of any individual test is unity. Therefore, we used the minimum $P$ value with Bonferroni correction to combine multiple tests when any individual test had $P=1$.

**Type I error evaluation.** To evaluate the type I error control of SAIGE-GENE and SAIGE-GENE+, we simulated binary phenotypes under the null hypothesis of no genetic effects based on the observed genotypes of 166,955 individuals of white British ancestry with WES data in UKBB (Supplementary Note). We conducted gene-based tests for 7,932 genes on the even chromosomes with missense and LoF variants using three different maximum MAF cutoffs (1%, 0.1% and 0.01%). In total, 158,640 gene-based tests were conducted for each maximum MAF cutoff for SAIGE-GENE and SAIGE-GENE+, respectively, and the quantile–quantile (Q–Q) plots are shown in Extended Data Fig. 4. Our simulation results suggest that SAIGE-GENE+ has well controlled type I error with case–control ratios of less than 1:100 when testing variants with maximum MAF 0.01% (Extended Data Fig. 4b).

We evaluated the type I error control of SAIGE-GENE, SAIGE-GENE+ and STAAR using the UKBB data (Fig. 1 and Supplementary Fig. 1). We applied these methods to four exemplary self-reported binary phenotypes with various case–control ratios in 166,955 individuals of white British ancestry with WES data to 18,372 genes, including all LoF and missense variants using three different maximum MAF cutoffs (1%, 0.1% and 0.01%). For STAAR, we used the relative coefficient cutoff of 0.05 for the sparse GRM to fit the null models.

**Power evaluation.** To evaluate the power of SAIGE-GENE+ and SAIGE-GENE, we simulated binary phenotypes based on genotypes of ten genes in 166,955 individuals of white British ancestry with WES data in UKBB (Supplementary Table 6). The selected genes showed significant gene–phenotype associations (Supplementary Table 5) and had a wide range of the number of rare variants from 2,901 (*APOB*) to 107 (*GPSM3*). The phenotype prevalence was set to be 10%, under which both SAIGE-GENE and SAIGE-GENE+ have well controlled type I error rates for Burden, SKAT and SKAT-O tests (Extended Data Fig. 3). Three scenarios with different settings of proportions of causal variants and magnitudes of effect sizes for causal variants were used: (1) low proportion of causal variants and small effect sizes, (2) low proportion of causal variants and large effect sizes, and (3) high proportion of causal variants and large effect sizes (Supplementary Table 7). More details about the simulation settings are described in the Supplementary Note. Our simulation results suggest that SAIGE-GENE+ has higher or similar empirical power than SAIGE-GENE (Supplementary Fig. 3 and Supplementary Table 8).

**UK Biobank WES data analysis.** We applied SAIGE-GENE+ to analyze 18,372 genes in the UKBB WES data from 166,955 white British individuals for 30 quantitative traits and 141 binary traits. UKBB protocols were approved by the National Research Ethics Service Committee, and participants signed written informed consent. Three different maximum MAF cutoffs (1%, 0.1% and 0.01%) and three different variant annotations (LoF only, LoF+missense and LoF+missense+synonymous) were applied, followed by aggregating multiple SKAT-O tests using the Cauchy combination[2,3] or minimum $P$ value for each gene. Variants were annotated using ANNOVAR[11]. The LoF variants include those annotated as frameshift deletion, frameshift insertion, nonframeshift deletion, nonframeshift insertion, splicing, stop gain and stop loss. Sex, age when attended assessment center and first four principal components estimated using all White British individuals were adjusted as covariates in all tests. A total of 250,656 markers with MAF ≥ 1%, which were pruned from the directly genotyped markers using the following parameters, were used to construct the GRM: window size of

500 base pairs (bp), step-size of 50 bp, and pairwise $r^2 < 0.2$. We used the relative coefficient cutoff of 0.05 for the sparse GRM for the variance ratio estimation after fitting the null models. The model was fitted with leave-one-chromosome-out (LOCO) to avoid proximal contamination.

**Computation cost evaluation.** Benchmarking was performed on randomly subsampled UKBB WES data (up to 150,000 individuals with white British ancestry) for glaucoma (1,741 cases and 162,408 controls). We reported the medians of five runs for run times and memory usage with samples randomly selected from the full sample set using five different sampling seeds. SAIGE-GENE and SAIGE-GENE+ use a two-step approach. Step 1 estimates the model parameters (that is, variance component and fixed effect coefficients) in the null model, and Step 2 conducts set-based association tests. SAIGE-GENE+ runs Step 1 with all covariates as offset, which leads to a decrease in the computation time (Supplementary Table 4). The computation cost of Step 1 in SAIGE-GENE+ is shown in Extended Data Fig. 8a and Supplementary Table 2. SAIGE-GENE+ has an option to use a sparse GRM to fit the null model in Step 1, which further reduces computation cost in Step 1 (Supplementary Note). Note that model parameters need to be estimated only once for each phenotype in Step 1 and can be used genome-wide regardless of MAF cutoffs and functional annotations in Step 2. We then compared computation time and memory usage of Step 2 (Fig. 2a, Supplementary Table 1 and Extended Data Fig. 7).

**Reporting summary.** Further information on research design is available in the Nature Research Reporting Summary linked to this article.

## Data availability

The PheWeb (v.0.9.15)-like visual server[10] and association summary statistics for 30 quantitative and 141 binary phenotypes of UKBB WES data analysis results are available at https://ukb-200kexome.leelabsg.org.

## Code availability

SAIGE-GENE+ is implemented as an open-source R package available at https://github.com/saigegit/SAIGE. SAIGE-GENE+ used in this study is deposited at https://zenodo.org/badge/latestdoi/470322837.

## References

11. Wang, K., Li, M. & Hakonarson, H. ANNOVAR: functional annotation of genetic variants from next-generation sequencing data. *Nucleic Acids Res.* **38**, e164 (2010).

## Acknowledgements

We thank A. Price for constructive comments and suggestions. We thank J. LeFaive for helping maintain the software to support the VCF and SAV input formats. This research was conducted using the UKBB Resource under application number 45227. S.L. was supported by Brain Pool Plus (BP+, Brain Pool+) Program through the National Research Foundation of Korea (NRF) funded by the Ministry of Science and ICT (2020H1D3A2A03100666, S.L). W.B. and Z.Z were supported by National Institutes of Health (NIH) R01 HG008773. W.Z. was supported by the National Human Genome Research Institute (NHGRI) of the NIH under award number T32HG010464 and K99HG012222. K.K.D. was supported by NIH/NHGRI award number K99HG012203. The research was supported by High-performance Computing Platform of Peking University.

## Author contributions

W.Z., W.B., Z.Z. and S.L. designed experiments. W.Z., W.B. and Z.Z. performed experiments and analyzed the UKBB data. W.Z. implemented the software with input from W.B. and Z.Z. K.K.D., K.A.J., K.J.K., B.M.N. and M.J.D. provided helpful advice. W.Z., W.B., Z.Z. and S.L. wrote the manuscript with input from all co-authors.

## Competing interests

B.M.N. is a member of Deep Genomics Scientific Advisory Board, has received travel expenses from Illumina, and also serves as a consultant for Avanir and Trigeminal solutions. K.J.K. is a consultant for Vor Biopharma. The remaining authors declare no competing interests.

## Additional information

**Extended data** is available for this paper at https://doi.org/10.1038/s41588-022-01178-w.

**Correspondence and requests for materials** should be addressed to Wei Zhou, Wenjian Bi or Seunggeun Lee.

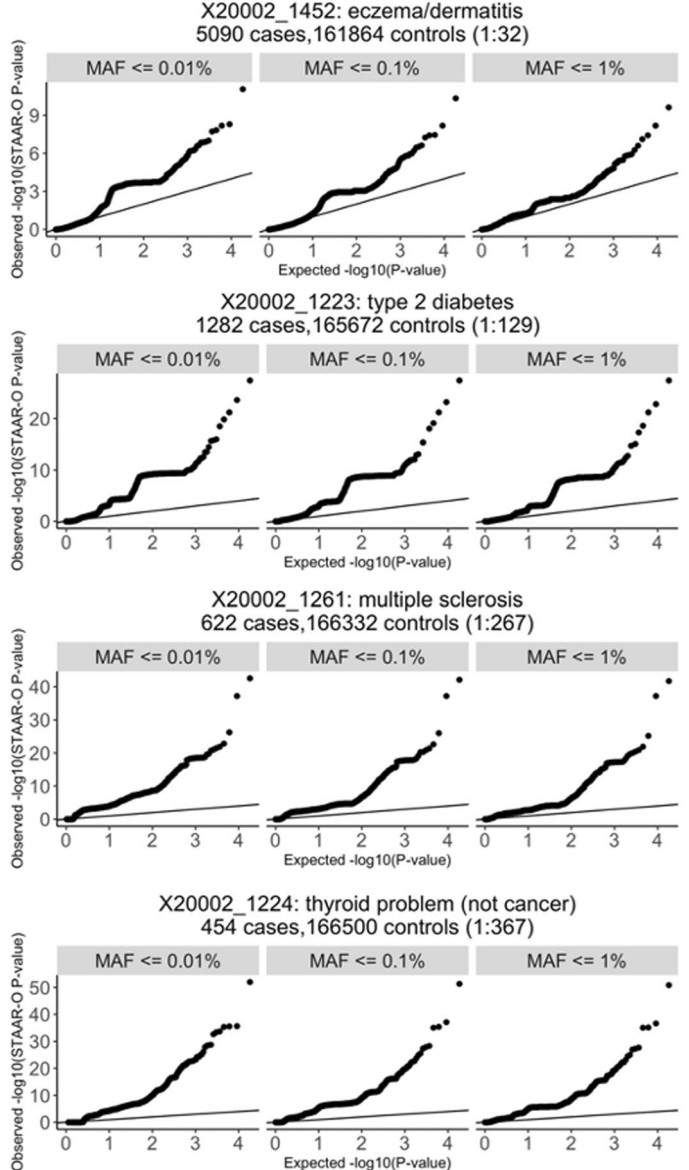

**Extended Data Fig. 1 | Quantile-quantile plots for STAAR-O tests *P* values for four exemplary binary phenotypes with different case–control ratios in the UKBB WES data.** The STAAR-O tests were performed for 18,372 genes with missense and loss-of-function (LoF) variants with three different maximum MAF cutoffs (1%, 0.1%, and 0.01%).

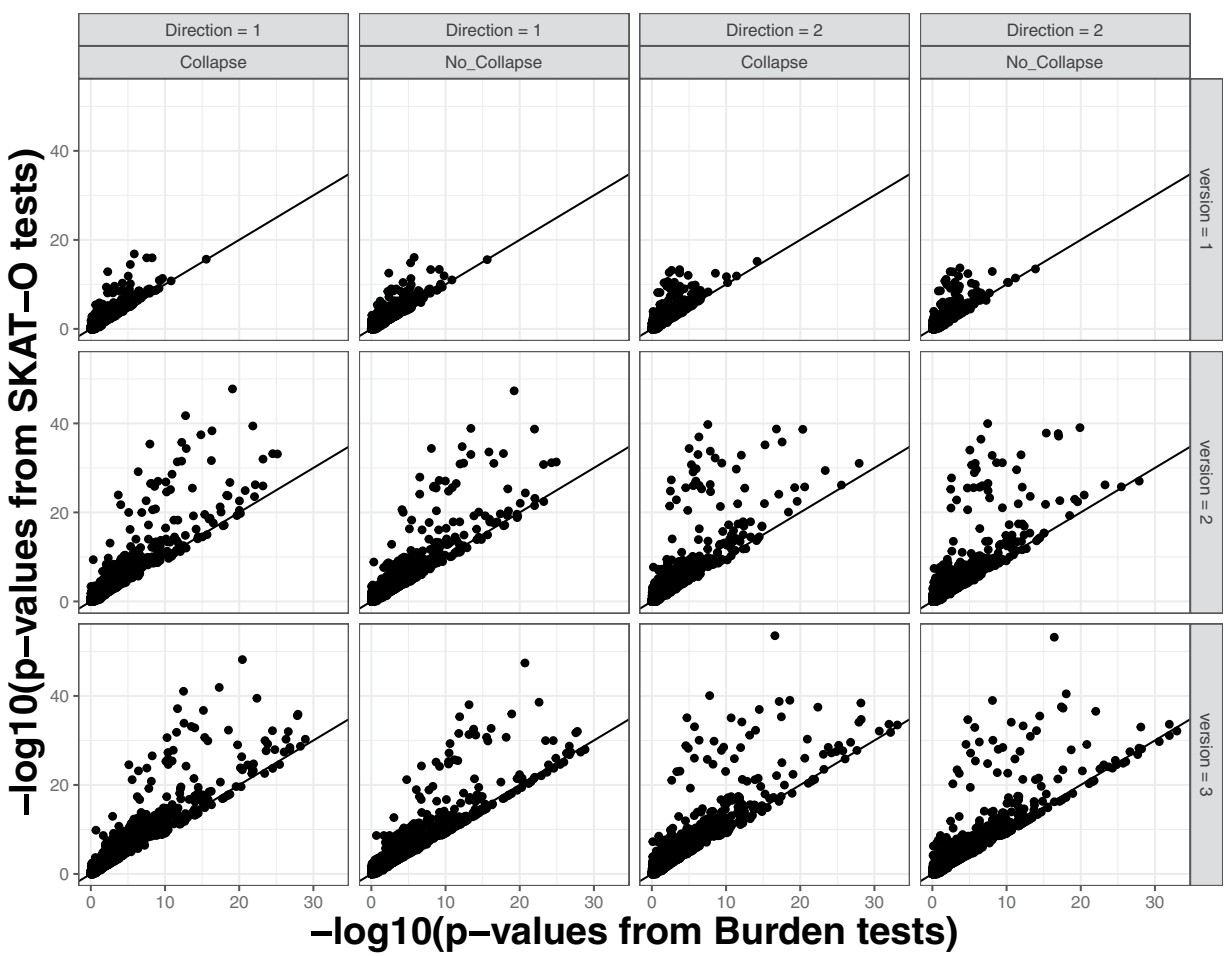

**Extended Data Fig. 2 | Scatter plots for association *P* values of SKAT-O and Burden tests in the simulation studies.** Each plot is based on test results for 1,000 test sets (100 data sets, each of which includes 10 genes; see Supplementary Table 6). The *x*-axis represents -$\log_{10}$ Burden test *P* values, and *y*-axis represents -$\log_{10}$ SKAT-O *P* values. The line in each plot represents the 45-degree line, so dots above the line have more significant *P* values from SKAT-O than the Burden test. The details of different simulation settings are presented in Supplementary Table 7. Tests conducted in the analysis were two-sided.

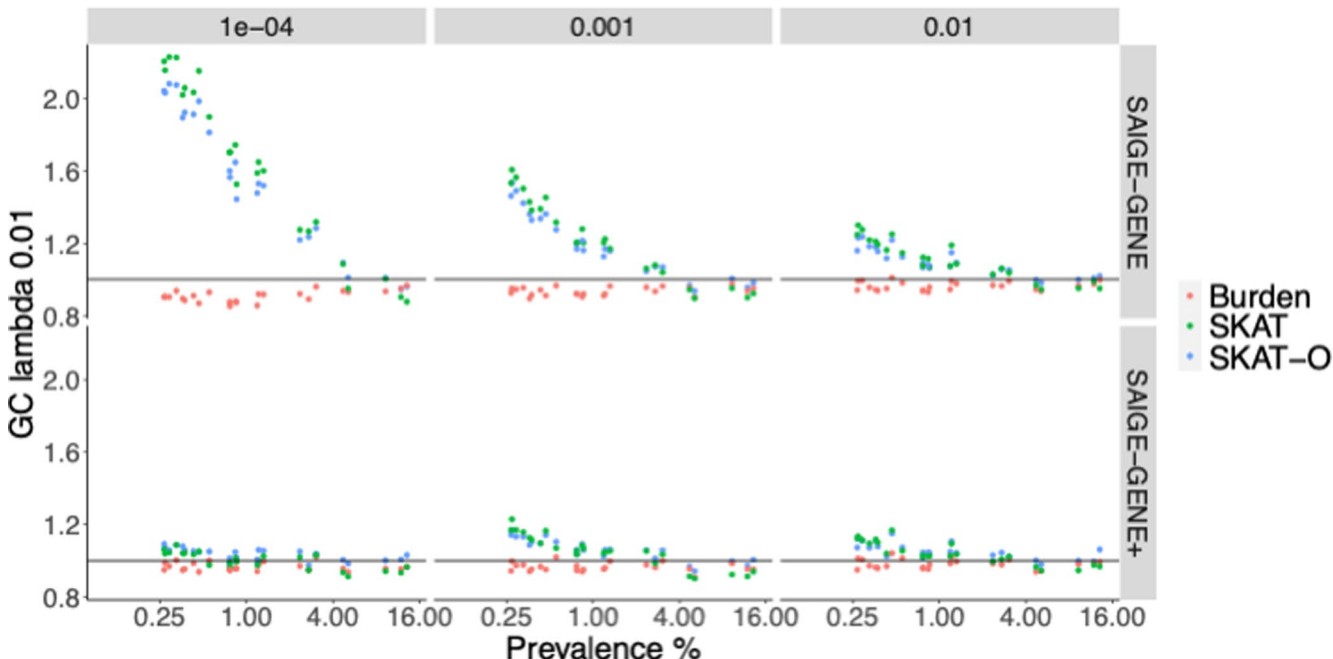

**Extended Data Fig. 3 | Genomic control inflation lambda values for 24 binary phenotypes in UKBB for SAIGE-GENE and SAIGE-GENE+.** Genomic control inflation lambda values based on the 1st percentile against the disease prevalence for 24 binary phenotypes in UKBB for SAIGE-GENE and SAIGE-GENE+ using three different maximum MAF cutoffs.

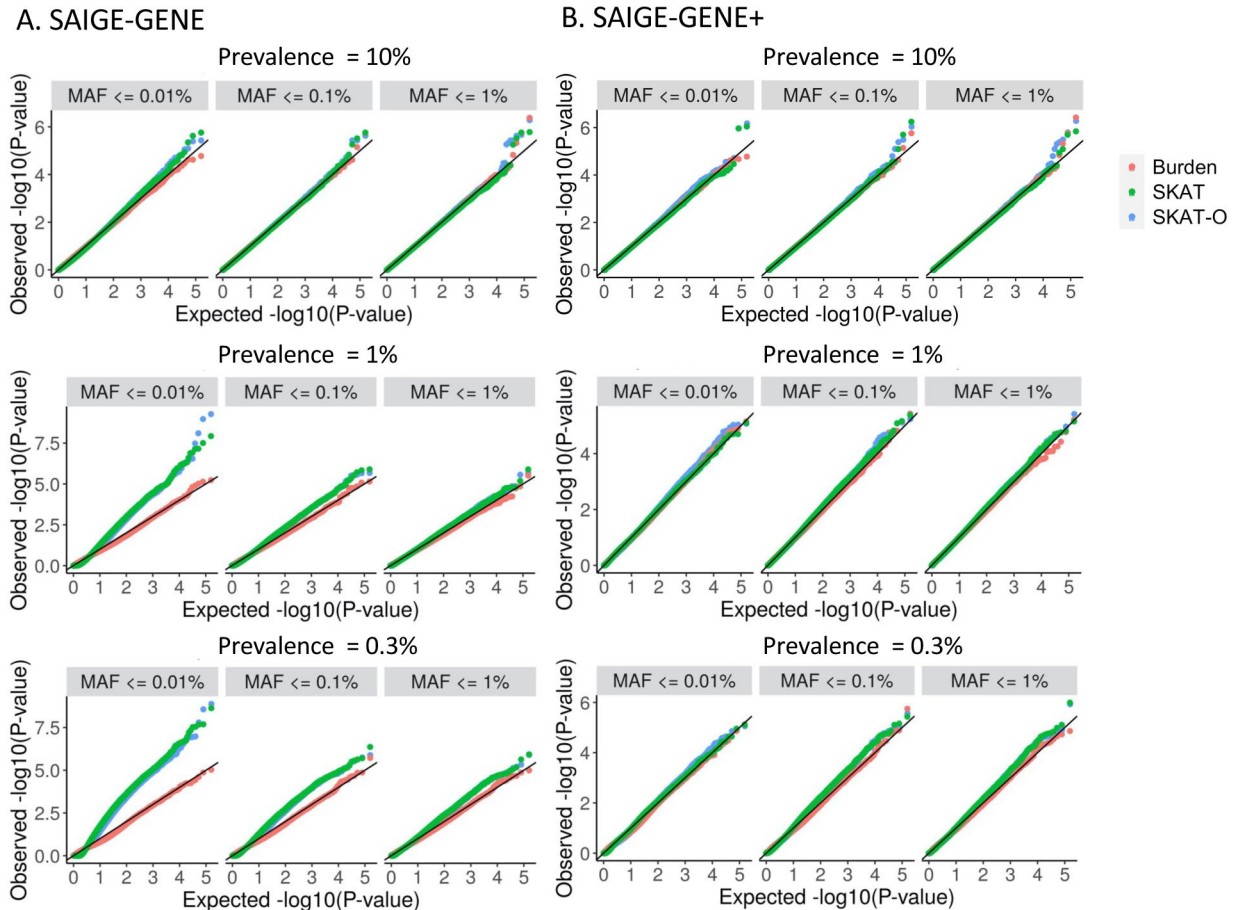

**Extended Data Fig. 4 | Quantile-quantile plots for Burden, SKAT, and SKAT-O tests *P* values for simulated phenotypes with prevalence 10%, 1%, and 0.3% based on the UKBB WES data under the null hypothesis. a**, Using SAIGE-GENE. **b**, Using SAIGE-GENE+, which collapses ultra-rare variants with MAC ≤10 prior to the gene-based association tests. The tests were performed for 18,372 genes with missense and loss-of-function variants with three different maximum MAF cutoffs (1%, 0.1%, and 0.01%). Tests conducted in the analysis were two-sided.

A. No per-marker weights are provided in the group file by the user. The weights of the collapsed variant and other non-ultra-rare variants (MAC > 10) are calculated based on their MAFs from Beta distribution $w_j = Beta(MAF_j, a_1, a_2)$. By default, $a_1 = 1, a_2 = 25$.

|  | Ultra-rare variants (MAC <= 10) | | | | | | | | | | Collapsed (max of dosages) |
|---|---|---|---|---|---|---|---|---|---|---|---|
| Sample 1 | 0 | 0 | 0 | 0 | 0 | 0 | 0 | 0 | 0 | 0 → | 0 |
| Sample 2 | 0 | 1 | 0 | 0 | 0 | 0 | 0 | 0 | 0 | 0 → | 1 |
| Sample 3 | 1 | 0 | 0 | 0 | 0 | 0 | 0 | 0 | 1 | 0 → | 1 |
| Sample 4 | 0 | 0 | 0 | 2 | 0 | 0 | 0 | 0 | 1 | 0 → | 2 |
| Sample 5 | 0 | 0 | 0 | 0 | 1 | 0 | 1 | 0 | 0 | 0 → | 1 |

*Weight of the collapsed variant*
$w = Beta(MAF, a_1, a_2)$

B. The per-marker weights are provided in the group file by the user

|  | Ultra-rare variants (MAC <= 10) | | | | | | | | | | Collapsed (max of the weighted dosages) |
|---|---|---|---|---|---|---|---|---|---|---|---|
| Sample 1 | 0 | 0 | 0 | 0 | 0 | 0 | 0 | 0 | 0 | 0 → | 0 |
| Sample 2 | 0 | 1 | 0 | 0 | 0 | 0 | 0 | 0 | 0 | 0 → | 0.2 |
| Sample 3 | 1 | 0 | 0 | 0 | 0 | 0 | 0 | 0 | 1 | 0 → | max(0.1, 0.9) = 0.9 |
| Sample 4 | 0 | 0 | 0 | 2 | 0 | 0 | 0 | 0 | 1 | 0 → | max(2*0.4, 0.9) = 0.9 |
| Sample 5 | 0 | 0 | 0 | 0 | 1 | 0 | 1 | 0 | 0 | 0 → | max(0.5, 0.7) = 0.7 |
| User-specified weight | 0.1 | 0.2 | 0.3 | 0.4 | 0.5 | 0.6 | 0.7 | 0.8 | 0.9 | 1 | |

**Extended Data Fig. 5 | Collapsing ultra-rare variants with MAC $\leq$ 10.** Demonstration on collapsing ultra-rare variants.

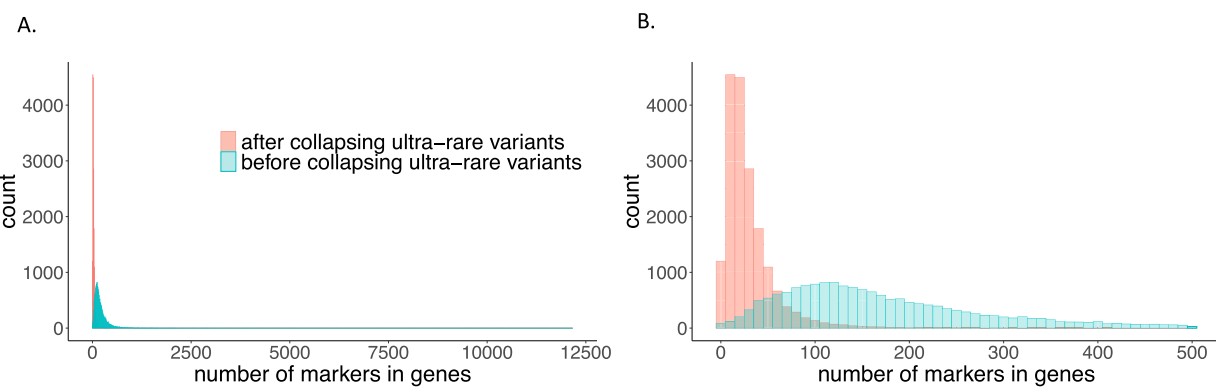

**Extended Data Fig. 6 | Histogram of number of genetic variants (missense and LoF) tested in each gene with maximum MAF 1% before and after collapsing the ultra-rare variants with MAC ≤ 10. a**, All genes. **b**, Genes with number of markers ≤ 500 before collapsing.

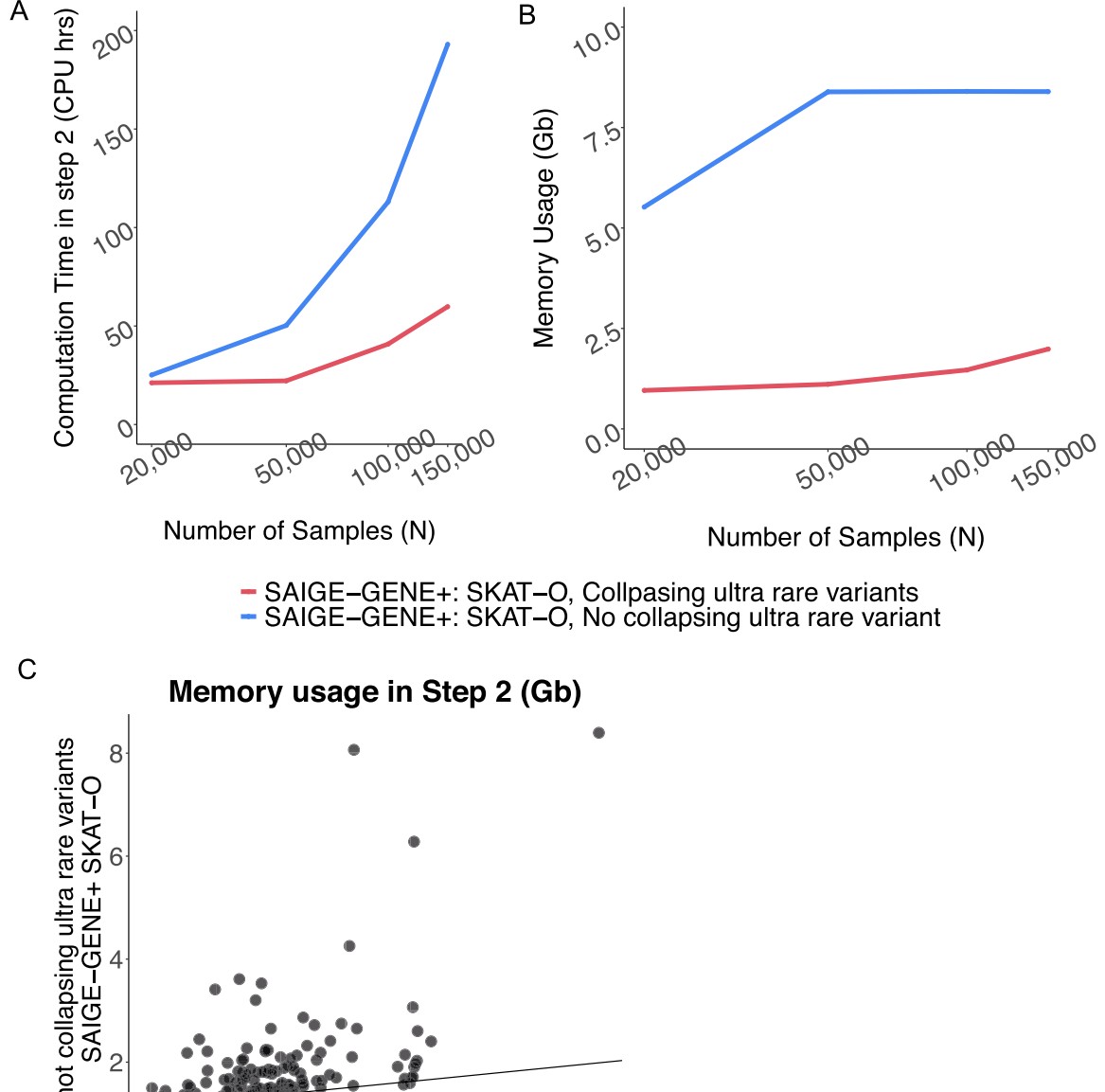

- ■ SAIGE−GENE+: SKAT−O, Collpasing ultra rare variants
- ■ SAIGE−GENE+: SKAT−O, No collapsing ultra rare variant

**Extended Data Fig. 7 | Computational cost of Step 2 in SAIGE-GENE+ with and without collapsing ultra-rare variants by sample sizes for gene-based tests for 18,372 genes with three maximum MAF cutoffs (1%, 0.1%, and 0.01%) and three variant annotations (LoF only, LoF + missense, and LoF + missense + synonymous).** In total, around 165,348 tests were run for each data set. Benchmarking was performed on randomly sub-sampled UK Biobank WES data with White British participants for glaucoma (1,741 cases and 162,408 controls). The reported run times and memory are medians of five runs with samples randomly selected from the full sample set using different sampling seeds. **a**, Plots of the time usage as a function of sample size ($N$). **b**, Plots of the maximum memory usage (for genes containing most variants) as a function of sample size ($N$). The x-axis is plotted on the $\log_2$ scale. **c**, Scatter plots of the memory usage when $N=150,000$ simulated with a random seed. We split the 165,348 tests into 133 chunks, each with ~150 genes. For each gene, nine SKAT-O tests were conducted corresponding to three different MAF cutoffs and functional annotations followed by combining the $P$ values using the Cauchy combination or minimum $P$-value approach. Tests conducted in the analysis were two-sided. Each dot in the plot is the maximum memory usage of a chunk among five runs with different random seeds.

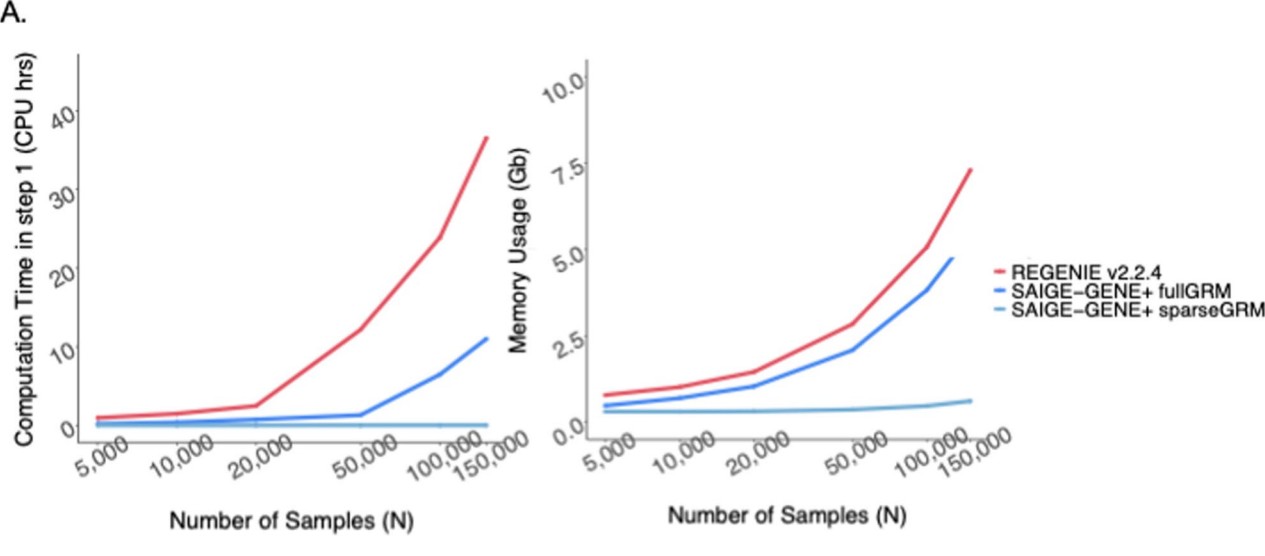

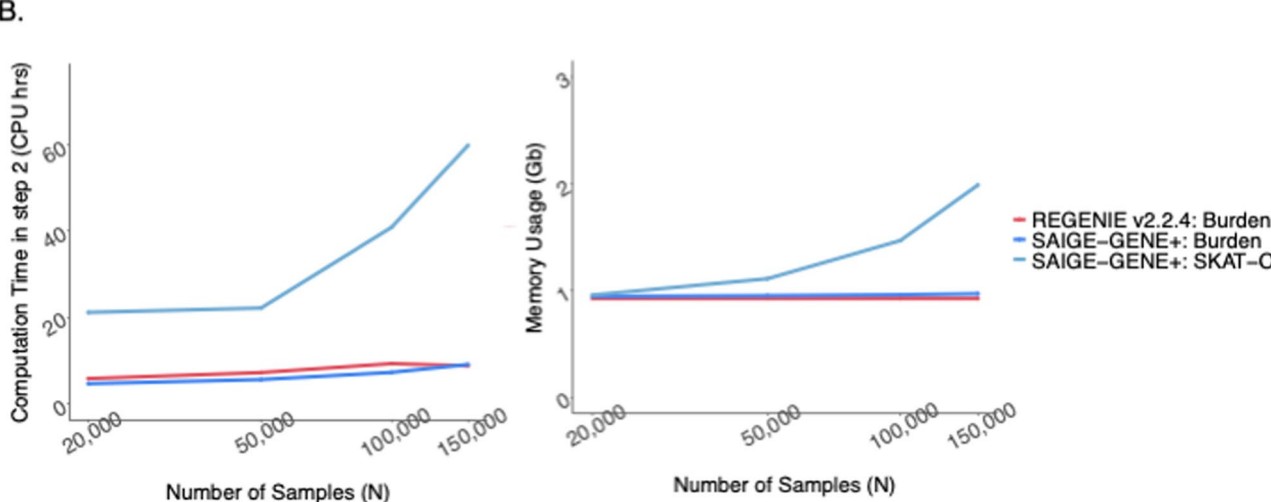

**Extended Data Fig. 8 | Computation cost in SAIGE-GENE+ and REGENIE2 by sample sizes for gene-based tests for 18,372 genes with three maximum MAF cutoffs (1%, 0.1%, and 0.01%) and three variant annotations (LoF only, LoF + missense, and LoF + missense + synonymous).** In total, 165,348 tests were run for each data set. Benchmarking was performed on randomly sub-sampled UK Biobank WES data with White British participants for glaucoma (1,741 cases and 162,408 controls). The reported run times and memory are medians of five runs with samples randomly selected from the full sample set using different sampling seeds. **a**, Pplots of the time usage and median memory usage in Step 1 as a function of sample size (*N*). **b**, Plots of the time usage and median memory usage in Step 2 as a function of sample size (*N*). Note that singletons only were also included as a mask in the Burden tests in both methods for a fair comparison. SAIGE-GENE+ further automatically output the *P* values by the Cauchy combination or minimum *P*-value approach. Tests conducted in the analysis were two-sided.

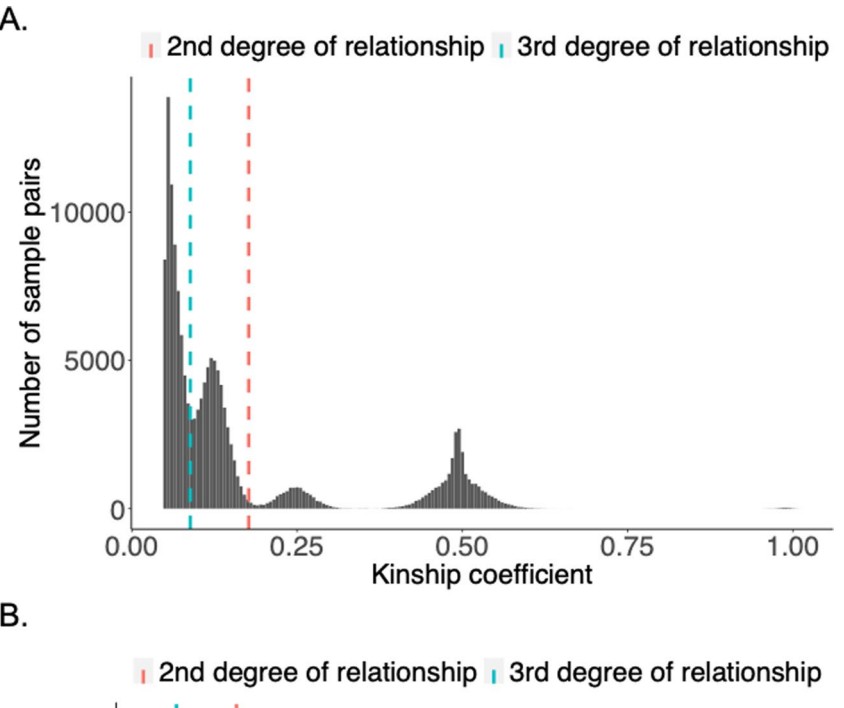

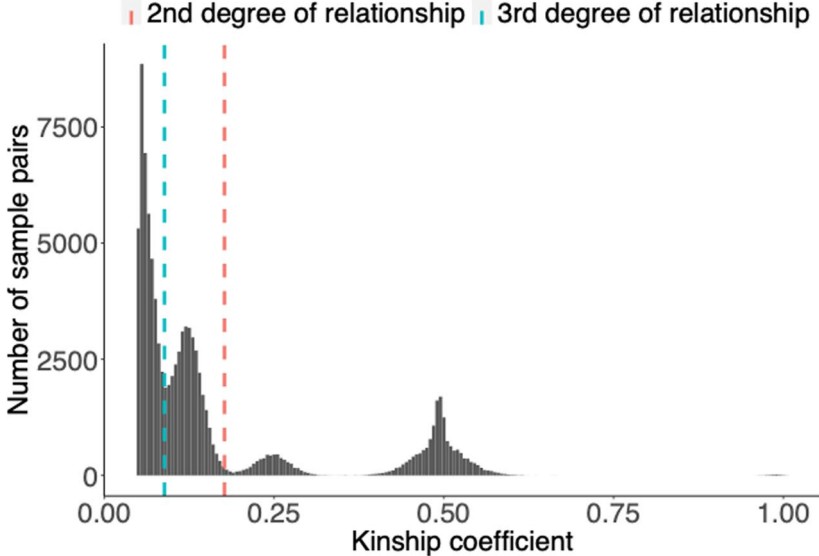

**Extended Data Fig. 9 | Histograms of kinship coefficients ($\geq$ 0.05) in UKBB. a**, All 408,910 samples. **b**, 200,643 samples with whole exome sequencing data available.

| | |
|---|---|

# Reporting Summary

## Statistics

For all statistical analyses, confirm that the following items are present in the figure legend, table legend, main text, or Methods section.

| n/a | Confirmed | |
|---|---|---|
| ☐ | ☒ | The exact sample size ($n$) for each experimental group/condition, given as a discrete number and unit of measurement |
| ☐ | ☒ | A statement on whether measurements were taken from distinct samples or whether the same sample was measured repeatedly |
| ☐ | ☒ | The statistical test(s) used AND whether they are one- or two-sided *Only common tests should be described solely by name; describe more complex techniques in the Methods section.* |
| ☐ | ☒ | A description of all covariates tested |
| ☐ | ☒ | A description of any assumptions or corrections, such as tests of normality and adjustment for multiple comparisons |
| ☐ | ☒ | A full description of the statistical parameters including central tendency (e.g. means) or other basic estimates (e.g. regression coefficient) AND variation (e.g. standard deviation) or associated estimates of uncertainty (e.g. confidence intervals) |
| ☐ | ☒ | For null hypothesis testing, the test statistic (e.g. $F$, $t$, $r$) with confidence intervals, effect sizes, degrees of freedom and $P$ value noted *Give P values as exact values whenever suitable.* |
| ☐ | ☒ | For Bayesian analysis, information on the choice of priors and Markov chain Monte Carlo settings |
| ☐ | ☒ | For hierarchical and complex designs, identification of the appropriate level for tests and full reporting of outcomes |
| ☐ | ☒ | Estimates of effect sizes (e.g. Cohen's $d$, Pearson's $r$), indicating how they were calculated |

*Our web collection on statistics for biologists contains articles on many of the points above.*

## Software and code

Policy information about availability of computer code

| Data collection | This research has been conducted using the UK Biobank Resource under application number 45227. |
|---|---|
| Data analysis | SAIGE-GENE(version0.44.6.1), https://github.com/weizhouUMICH/SAIGE/; SAIGE-GENE+ (version1.0.0), https://github.com/saigegit/SAIGE; STAAR (version0.9.5), https://github.com/xihaoli/STAAR; REGENIE(version2.2.4) https://github.com/rgcgithub/regenie. |

For manuscripts utilizing custom algorithms or software that are central to the research but not yet described in published literature, software must be made available to editors and reviewers. We strongly encourage code deposition in a community repository (e.g. GitHub). See the Nature Portfolio guidelines for submitting code & software for further information.

## Data

Policy information about availability of data

All manuscripts must include a data availability statement. This statement should provide the following information, where applicable:

- Accession codes, unique identifiers, or web links for publicly available datasets
- A description of any restrictions on data availability
- For clinical datasets or third party data, please ensure that the statement adheres to our policy

SAIGE-GENE+ is implemented as an open-source R package available at https://github.com/saigegit/SAIGE. The version used in the paper has been deposited at https://zenodo.org/badge/latestdoi/470322837. The summary statistics and QQ plots for 30 quantitative phenotypes and 141 binary phenotypes in UK Biobank by SAIGE-GENE+ are currently available for public download at https://storage.googleapis.com/leelabsg/saige-gene/reformat_all_withPhenoDetails.txt.

# Field-specific reporting

Please select the one below that is the best fit for your research. If you are not sure, read the appropriate sections before making your selection.

☒ Life sciences          ☐ Behavioural & social sciences          ☐ Ecological, evolutionary & environmental sciences

For a reference copy of the document with all sections, see nature.com/documents/nr-reporting-summary-flat.pdf

# Life sciences study design

All studies must disclose on these points even when the disclosure is negative.

| | |
|---|---|
| Sample size | We analyzed publicly available UK Biobank whole exome sequencing data of white British samples (sample size=166,955). For the study design, please refer UK Biobank (http://www.ukbiobank.ac.uk/). |
| Data exclusions | To avoid confounding, analyses were done in white British samples in UK Biobank with non White British samples excluded. |
| Replication | For all exome-wide significant loci, we searched published GWAS studies to check whether they were known (replicated) or novel. |
| Randomization | We used publicly available UK Biobank data to illustrate the performance of the method. We randomly selected subsets of samples from the data set to evaluate method performance. |
| Blinding | We used coded public data, and hence were blinded. |

# Reporting for specific materials, systems and methods

We require information from authors about some types of materials, experimental systems and methods used in many studies. Here, indicate whether each material, system or method listed is relevant to your study. If you are not sure if a list item applies to your research, read the appropriate section before selecting a response.

### Materials & experimental systems

| n/a | Involved in the study |
|---|---|
| ☒ | ☐ Antibodies |
| ☒ | ☐ Eukaryotic cell lines |
| ☒ | ☐ Palaeontology and archaeology |
| ☒ | ☐ Animals and other organisms |
| ☐ | ☒ Human research participants |
| ☒ | ☐ Clinical data |
| ☒ | ☐ Dual use research of concern |

### Methods

| n/a | Involved in the study |
|---|---|
| ☒ | ☐ ChIP-seq |
| ☒ | ☐ Flow cytometry |
| ☒ | ☐ MRI-based neuroimaging |

## Human research participants

Policy information about studies involving human research participants

| | |
|---|---|
| Population characteristics | The UK Biobank study population (http://www.ukbiobank.ac.uk/) is residents of the UK aged 40-69 years at recruitment and living within a reasonable travelling distance of an assessment centre. |
| Recruitment | The UK Biobank participants were selected using the NHS register, and invited to volunteer for the study. Recruitment was carried out between 2007 and 2010. Full details of the recruitment process are available in reference (UK Biobank: Protocol for a large-scale prospective epidemiological resource, 2007). |
| Ethics oversight | National Research Ethics Service Committee (UK Biobank) |

Note that full information on the approval of the study protocol must also be provided in the manuscript.

