## [Peer Review File · Nature Genetics]

Peer Review Information

Manuscript Title: SAIGE-GENE+ improves the efficiency and accuracy of set-based rare variant association tests

Corresponding author name(s): Seunggeun Lee

Reviewer Comments & Decisions:

Decision Letter, initial version:

29th November 2021

Dear Seunggeun,

Your Brief Communication "Set-based rare variant association tests for biobank scale sequencing data sets" has been seen by three referees. You will see from their comments below that, while they find your work of potential interest, they have raised substantial concerns that must be addressed. In light of these concerns, we cannot accept the manuscript for publication at this time, but we would be interested in considering a substantially revised version that addresses the referees' criticisms.

We hope you will find the referees' comments useful as you decide how to proceed. If you wish to submit a substantially revised manuscript, please bear in mind that we will be reluctant to approach the referees again in the absence of major revisions.

To guide the scope of the revisions, the editors discuss the referee reports within the team, including with the chief editor, with a view to identifying key priorities that should be addressed in revision, and sometimes overruling referee requests that are deemed beyond the scope of the current study. In this case, we specifically ask that you perform additional power calculations as requested by Reviewer #1, improve the software coding and documentation as requested by Reviewer #2, explain how functional variant weights are collapsed for ultra-rare variants as requested by Reviewer #2, clarify how the software update achieves performance gains for different variant frequency thresholds as requested by Reviewer #3, and provide specific comparisons between this updated SAIGE-GENE software package and REGENIE as requested by Reviewers #2 and #3. We hope you will find this prioritized set of referee points to be useful when revising your study. Please do not hesitate to get in touch if you would like to discuss these issues further.

If you choose to revise your manuscript taking into account all reviewer and editor comments, please

highlight all changes in the manuscript text file. At this stage, we will need you to upload a copy of the manuscript in MS Word .docx or similar editable format.

*2) If you have not done so already please begin to revise your manuscript so that it conforms to our Brief Communication format instructions, available [here](http://www.nature.com/ng/authors/article_types/index.html). Refer also to any guidelines provided in this letter.

[REDACTED]

If you wish to submit a suitably revised manuscript we would hope to receive it within 3-6 months. If you cannot send it within this time, please let us know. We will be happy to consider your revision so long as nothing similar has been accepted for publication at Nature Genetics or published elsewhere. Should your manuscript be substantially delayed without notifying us in advance and your article is eventually published, the received date would be that of the revised, not the original, version.

Nature Genetics is committed to improving transparency in authorship. As part of our efforts in this

2direction, we are now requesting that all authors identified as 'corresponding author' on published papers create and link their Open Researcher and Contributor Identifier (ORCID) with their account on the Manuscript Tracking System (MTS), prior to acceptance. ORCID helps the scientific community achieve unambiguous attribution of all scholarly contributions. You can create and link your ORCID from the home page of the MTS by clicking on 'Modify my Springer Nature account'. For more information please visit www.springernature.com/orcid.

Thank you for the opportunity to review your work.

Sincerely,
Kyle

Kyle Vogan, PhD
Senior Editor
Nature Genetics
<https://orcid.org/0000-0001-9565-9665>

Referee expertise:

Referee #1: Genetics, statistics, computational methods

Referee #2: Genetics, computational methods, statistics

Referee #3: Genetics, rare variants

Reviewers' Comments:

Reviewer #1:
Remarks to the Author:

The SAIGE-GENE paper was published last year with the claim that it was scalable to large datasets and also accounted for case-control imbalance. The current paper suggests that SAIGE-GENE was not scalable (especially to large genes) and it does not properly account for case-control imbalance.

In this new paper the authors suggest to solve the issue by collapsing rare variants into a single marker. This is a very simple thing to do. I am not sure it meets the bar for a substantial new method. It is also a change to the assumptions of the model, as rare variants are now assumed to all act in the same direction. No power evaluations are carried out to evaluate this change in the model.

Reviewer #2:

Remarks to the Author:

In this manuscript, the authors describe SAIGE-GENE+, a minor update to their well-known SAIGE software package. The new update performs collapsing for very rare variants, which speeds up calculations and better controls error rates.

This note presents a welcome and much-awaited improvement to SAIGE-GENE. As the authors admit, previous versions suffered from significant computational and error control issues when analysing rare variants in biobank-scale datasets. The authors introduce a simple procedure to address both problems at once: collapse very rare genotypes even when SKAT tests are performed. While a software update usually would not warrant publication in a high-impact journal, the potential scope of this new technique and the dearth of methods to perform weighted, optimal tests on biobank-sized datasets provides sufficient justification in my eyes.

I have tested SAIGE-GENE+ and have confirmed that it works well. On a test on 1000 chunks of UK Biobank WES, everything else being equal, there was an average reduction of 4000 seconds in runtime per chunk when using the collapsing scheme versus not using it. The genomic inflation factor changed from 1.26 to 1.03. Similar results were observed when changing the upper MAF threshold from 1% to 0.1%.

Major concerns remain about this piece of software, however, that should be addressed before this update is published.

Major comments:

1) The title does not reflect the small magnitude of the changes implemented in SAIGE-GENE. I would recommend changing it to something along the lines of "An update to SAIGE-GENE improves computational performance and error rates for gene-based tests in biobank-scale sequencing datasets". Please also consider dropping the moniker "SAIGE-GENE+". This is not a separate package, rather, the authors have added 1 argument to their existing package, which does not justify such rebranding.

2) This paper would be the third publication about SAIGE, after the original single-point method and the subsequent SAIGE-GENE paper. Yet the software feels like it is still very much in beta stage. The documentation is clunky, the wiki has dead links, and the package content is littered with debug code. Some data objects are pasted to stdout, there are messages such as "DEBUG1" "DEBUG2" "DEBUG3", etc. Options naming is inconsistent particularly for binary options and flags (--IsBinaryOption TRUE and --BinaryOption TRUE exist). If this software is to be considered as mature, it needs to look like one, with consistent options naming, a robust and detailed documentation, a log argument and a verbosity argument. There hasn't been a release on github since version 0.29.5, please add subsequent version releases up until 0.44.6.5. Provide documentation and a suitable user-friendly interface to the CCT function(perhaps rename it also).

3) There are currently 149 open issues on the github page, many of which have not been answered.

4There is obviously a lot of interest in this software, yet the authors seem to struggle to provide support. The authors should solve and close as many issues as possible before releasing a published update. I also note that many closed issues did not result in code changes such as, for example, to provide more informative error messages.

4) One of the main advantage of SAIGE-GENE compared to its more performant competitors like REGENIE2 is that SAIGE-GENE allows to perform optimal tests (SKAT-O) and allows functional weights. Yet the paper does not explain how functional variant weights are collapsed when ultra-rare variants are collapsed as part of this scheme. Please add a paragraph detailing how this is done.

Minor Comments

1) line 46-47, add here that SAIGE-GENE performs SKAT-(O) tests which are agnostic to genetic architecture. It is also the only one performing weighted tests at scale. Mention and compare the big competition of SAIGE-GENE, which is REGENIE2.

2) l. 115. remove "the" age at menopause.

3) l. 116 remove "the" lymphocyte count.

4) l. 126 do not use "insignificant". Use "non-significant" or "miss the p-value threshold" or similar.

5) l.132 conducted "on" LoF variants, not "to".

Reviewer #3:

Remarks to the Author:

The authors present an improvement upon SAIGE-GENE that will be useful to the community. The evidence showing faster computational times and reduction of test statistic inflation when variants are very rare is really helpful.

However, I felt the paper was lacking details in why and how SAIGE-GENE+ performed better than SAIGE-GENE, as in what changed about the methods to create such an improvement. I noticed much focus on the 'ultra-rare' variant analysis with $MAC < 10$, but that doesn't explain why things performed better for $MAC < 0.1\%$ cutoff as well.

I also didn't understand why the p-value improved for BRCA from $x10^{-3}$ to $x10^{-8}$ when the same MAF cutoff was used; same for GCK, as here: "For example, BRCA2 for breast cancer with MAF 0.1% has p-value 7.62×10^{-8} in SAIGE-GENE+ and 1.65×10^{-3} 96 in SAIGE-GENE, and GCK for diabetes with MAF 0.1% p-value 1.22×10^{-13} in SAIGE-GENE+ and 4.06×10^{-6} 97 in SAIGE-GENE." Why would the p-value improve if the MAF cutoff was the same?

There was also a glaring omission of regenie in this manuscript. regenie, which was based on SAIGE,

5has really arisen as the key gene-based collapsing analysis method this past year, and a comparison to regenie would be really show how SAIGE-GENE+ is an improvement.

Author Rebuttal to Initial comments

Response to Reviewers Comments

Title: Set-based rare variant association tests for biobank scale sequencing data sets

We thank the reviewers and editors for their thoughtful and constructive comments that helped improve the manuscript. Below are our detailed responses (in bold). Quoted text from the manuscript is highlighted in blue. To distinguish comments and responses, comments are italicized. In the main manuscript and supplementary materials, added or revised text is highlighted in yellow.

Editors' Comments:

To guide the scope of the revisions, the editors discuss the referee reports within the team, including with the chief editor, with a view to identifying key priorities that should be addressed in revision, and sometimes overruling referee requests that are deemed beyond the scope of the current study. In this case, we specifically ask that you perform additional power calculations as requested by Reviewer #1, improve the software coding and documentation as requested by Reviewer #2, explain how functional variant weights are collapsed for ultra-rare variants as requested by Reviewer #2, clarify how the software update achieves performance gains for different variant frequency thresholds as requested by Reviewer #3, and provide specific comparisons between this updated SAIGE-GENE software package and REGENIE as requested by Reviewers #2 and #3. We hope you will find this prioritized set of referee points to be useful when revising your study. Please do not hesitate to get in touch if you would like to discuss these issues further.

RE: Thank you for providing us with the prioritized set of points. We have tried to address all of them. Please see below. More detailed explanations can be found in our response to each reviewer's comment.

Point 1. Perform additional power calculations

RE: We carried out power simulation studies using UK-Biobank WES genotype data and different scenarios of the proportion of the causal variants, effect sizes, and effect direction (Supplementary Table 6). In particular, we simulated binary phenotypes based on the observed genotypes of ten genes in the WES data of 166,955 samples with white British ancestry (Supplementary Table 5). The selected genes showed significant gene-phenotype associations (Supplementary Table 4) and had wide ranges of the number of rare variants from 2901 (*APOB*) to 107 (*GPSM3*) (Supplementary Table 5). Three scenarios with different settings of proportions of causal variants across the multiple functional annotations and magnitudes of effect sizes were used: 1) low proportion of causal variants and low effect sizes, 2) low proportion of causal variants and high effect sizes, and 3) high proportion of causal variants and high effect sizes (Supplementary Table 6).

Our results showed that 1) testing multiple variant sets (multiple masks) in SAIGE-GENE+ had higher empirical power than our previous SAIGE-GENE that only considered one variant set (single mask) (Supplementary Figure 10). 2) Collapsing ultra-rare variants did not hurt power even in the presence of variants with different association directions (Supplementary Figure 8, Supplementary Table 7). 3) Burden test approach implemented in SAIGE-GENE+ and REGENIE2 performed comparably (Supplementary Figure 9), both of which have lower power compared to SKAT-O (Supplementary Figure 2).

Point 2. Improve the software coding and documentation

RE: We thoroughly investigated the software code, removed any redundant code, and debug outputs. We also converted many parts of R code to C++ code to further improve the computation performance. Details on the code improvement can be found at Supplementary Note F. We have created a new github site with clear documentation (<https://saigegit.github.io/SAIGE-doc/>).

7Point 3. Explain how functional variant weights are collapsed for ultra-rare variants

RE: We have added a section in Method (the first paragraph in Method) and a figure (Supplementary Figure 15) to explain how the functional variant weights are collapsed for ultra-rare variants. In the presence of the functional weights provided by a user, we first calculate weighted dosage by multiplying the weight to each genotype and then use the maximum value as the collapsed dosage.

Point 4. Clarify how the software update achieves performance gains for different variant frequency thresholds

RE: We have added a paragraph in Results to explain why different variant frequency thresholds can achieve power improvement. One example of a more significant p-value by MAF thresholding is the association between the gene *PDCD1LG* and chronic lymphoid leukemia. The association is more significant with a lower MAF cutoff than $MAF \leq 1\%$. The marker-level associations and the associations with each MAF cutoff and functional annotation mask can be found at the browser <https://ukb-200kexome.leelabsg.org/assoc/PDCD1LG2/204.12>. The underlying reason for this observation could be that the associations are largely enriched in the rarer variants in the gene, e.g. the most significant variant is the low frequency variant rs7854303 (9:5557672, $MAF=3.4 \times 10^{-4}$). Using a higher MAF cutoff 1% will include many non-causal variants, which decreases the power of the set-based association tests. Given that the underlying genetic architecture is usually unknown for different genes and phenotypes, incorporating multiple MAF cutoffs and functional annotations would lead to a higher power than using a single MAF cutoff or functional annotation. We have also conducted simulation studies to illustrate it. The plots showing p-value comparisons are presented in Supplementary Figure 10, and the medians of the Chi-square test statistics are included in Supplementary Table 7.

Point 5. Provide specific comparisons between this updated SAIGE-GENE software package and REGENIE

RE: We compared the updated SAIGE-GENE+ software to REGENIE2 ver. 2.2.4. Please note that REGENIE2 implemented the Burden test only. As expected, the Burden test p-values from SAIGE-GENE+ and REGENIE2 were highly concordant (Pearson's correlation $R^2 = 0.99$ for $-\log_{10}(\text{p-value})$) (Supplementary Figure 9).

Next, we compared the computation cost. Both SAIGE-GENE+ and REGENIE2 require two steps, Step 1 of fitting the null model and Step2 testing for associations for each gene, so we measure the computation cost separately. In step1, we have found that our software achieves shorter computation time and lower memory usages than REGENIE2 (Supplementary Figure 7A). The computation cost gap was much larger when the sparse GRM was used in SAIGE-GENE+ (SAIGE GENE+: <1 CPU min, and 0.61Gb vs REGENIE2: 36.5 CPU hours and 7.3 Gb). In Step 2, similar computation cost was observed for the two methods for Burden tests (Supplementary Figure 7B and Supplementary Table 3): 8.8 CPU hours and 0.93Gb by REGENIE2 and 9.1 CPU hours and 0.97Gb by SAIGE-GENE+. While the SKAT-O tests in SAIGE-GENE+ required nearly 6x more computation time (60 CPU hours) and 2x more memory (2.0 Gb) (Supplementary Figure 7B and Supplementary Table 3), through simulation studies, we observed that SKAT-O tests had higher power than Burden tests in all different scenarios (Supplementary Table 6) with more significant p-values (Supplementary Figure 2) and higher mean Chi-square statistics (Supplementary Table 7). We now have added a section in Supplementary Note (Supplementary Note D) to summarize the comparison with REGENIE.

When we completed this revision, REGENIE3 was just released with gene-based tests, including SKAT and SKAT-O. From the document, they basically implemented our approach (SAIGE-GENE+) with collapsing ultra rare variants with $\text{MAC} \leq 10$ and applying saddlepoint approximation to adjust for case-control imbalance. We did a quick evaluation using UKBB WES and found that REGENIE3 resulted in greatly inflated QQ plots (Figure 1 shown below). Since REGENIE3 is a software package implementing existing approaches, including SKAT and SKAT-O tests in SAIGE-GENE+, and the code may not be extensively tested as shown in type I error inflation, we didn't include REGENIE3 comparison in our manuscript.

Figure 1. The data set contains 150,000 samples randomly sub-sampled from UK Biobank WES data with White British participants. SKAT-O tests were performed for glaucoma (1,606 cases and 148,394 controls) using REGENIE3 and SAIGE-GENE+ for 18,372 genes with three maximum MAF cutoffs: 1%,

0.1%, and 0.01% and three variant annotations: LoF only, LoF + missense, and LoF + missense +synonymous.

Reviewers' Comments:

Reviewer #1:

Remarks to the Author:

The SAIGE-GENE paper was published last year with the claim that it was scalable to large datasets and also accounted for case-control imbalance. The current paper suggests that SAIGE-GENE was not scalable (especially to large genes) and it does not properly account for case-control imbalance.

In this new paper the authors suggest to solve the issue by collapsing rare variants into a single marker. This is a very simple thing to do. I am not sure it meets the bar for a substantial new method. It is also a change to the assumptions of the model, as rare variants are now assumed to all act in the same direction. No power evaluations are carried out to evaluate this change in the model.

RE: We thank the reviewer's comment. At the time when SAIGE-GENE was developed, the 200K UK Biobank WES data were not publically available yet, so it was evaluated using the imputed data focusing on the performance for accounting for sample relatedness and case-control imbalance while handling large sample sizes. More recently, the release of the 200K WES UK Biobank data provides the opportunity to test associations of genes or regions with lower MAF cutoffs and multiple functional annotations. But challenges were then observed, including the inflated type I errors for SKAT and SKAT-O tests when lower maximum MAF cutoffs, such as 0.1% and 0.01%, were used for phenotypes with more unbalanced case control ratios (Supplementary Figure 3) and SAIGE-GENE was not computational efficient enough to handle multiple testing masks (different MAF cutoffs and function annotations).

We agree that collapsing ultra-rare variants with $MAC \leq 10$ assumes the same effect direction of the variants. But this approach has several important advantages, including leading to better controlled type I errors in SKAT and SKAT-O tests and reducing the computation cost, which then makes the program computationally feasible for applying multiple testing masks on each gene or region, such as

11different functional annotations and maximum MAF cutoffs. In line with the suggestion, we have conducted extensive simulation studies to evaluate the power of SAIGE-GENE+ and the original SAIGE-GENE using various scenarios with different proportions of causal variants and proportions with the same effect direction and magnitudes of effect sizes (Supplementary Table 6). In all simulation scenarios, collapsing ultra-rare variants gives similar or greater power than the previous version of SAIGE-GENE (Supplementary Figure 8, Supplementary Table 7). We have described the simulation study results in the Main section.

The details of the simulation studies for power evaluation are described in the subsection “Power evaluation” in the Methods section and Supplementary Note A. Results have been described in the Main section “We evaluated the powers of SAIGE-GENE+ and SAIGE-GENE through extensive simulation studies based on the real genotypes of 10 genes in the UKBB WES data (Supplementary Table 5, Supplementary Note A, Methods) for binary traits. Three scenarios with different settings of proportions of causal variants across the multiple functional annotations and different settings of absolute effect sizes for causal variants were used. For each scenario, two settings of effect directions were used: 1. All causal variants increased disease risk; 2. 100% of LoF, 80% of missense, and 50% of synonymous causal variants increased disease risk and the other causal variants decreased disease risk (Supplementary Table 6). The prevalence of the binary phenotypes was set to be 10%, under which both SAIGE-GENE and SAIGE-GENE+ have well controlled type I error rates for Burden, SKAT, and SKAT-O tests. We observed that in all scenarios, SAIGE-GENE+ had higher or similar empirical power than SAIGE-GENE (Supplementary Figure 8) with the increased median Chi-square statistics (Supplementary Table 7). Our results also demonstrated as previously reported that the SKAT-O test can have higher power than the Burden test with more significant p-values and higher median Chi-square statistics (Supplementary Figure 2, Supplementary Table 7), while the Burden test p-values from SAIGE-GENE+ are highly concordant to p-values from REGENIE2 (Pearson’s correlation $R^2 = 0.99$ for $-\log_{10}(\text{p-value})$) (Supplementary Figure 9). In addition, the simulation results suggested incorporating multiple functional annotations (LoF, LoF+Missense, LoF+Missense+Synonymous) and maximum MAF cutoffs (0.01%, 0.1%, and 1%) can have an increased power compared to only using a single maximum MAF cutoffs (1%) on one set of function annotation (LoF+Missense+Synonymous) (Supplementary Figure 10, Supplementary Table 7).”

Reviewer #2:

Remarks to the Author:

In this manuscript, the authors describe SAIGE-GENE+, a minor update to their well-known SAIGE software package. The new update performs collapsing for very rare variants, which speeds up calculations and better controls error rates.

This note presents a welcome and much-awaited improvement to SAIGE-GENE. As the authors admit, previous versions suffered from significant computational and error control issues when analysing rare variants in biobank-scale datasets. The authors introduce a simple procedure to address both problems at once: collapse very rare genotypes even when SKAT tests are performed. While a software update usually would not warrant publication in a high-impact journal, the potential scope of this new technique and the dearth of methods to perform weighted, optimal tests on biobank-sized datasets provides sufficient justification in my eyes.

I have tested SAIGE-GENE+ and have confirmed that it works well. On a test on 1000 chunks of UK Biobank WES, everything else being equal, there was an average reduction of 4000 seconds in runtime per chunk when using the collapsing scheme versus not using it. The genomic inflation factor changed from 1.26 to 1.03. Similar results were observed when changing the upper MAF threshold from 1% to 0.1%.

We thank the reviewer for this comprehensive summary and testing SAIGE-GENE+ with UK Biobank WES data.

Major concerns remain about this piece of software, however, that should be addressed before this update is published.

Major comments:

131) *The title does not reflect the small magnitude of the changes implemented in SAIGE-GENE. I would recommend changing it to something along the lines of "An update to SAIGE-GENE improves computational performance and error rates for gene-based tests in biobank-scale sequencing datasets". Please also consider dropping the moniker "SAIGE-GENE+". This is not a separate package, rather, the authors have added 1 argument to their existing package, which does not justify such rebranding.*

RE: We thank the reviewer for the suggestion. We contemplated it. However, we prefer to call the program SAIGE-GENE+ to distinguish it from all previous versions of SAIGE-GENE. In addition to the ultra-rare variant collapsing, the new approach also includes the Cauchy combination to aggregate tests from different variant sets, so the new name also refers to this.

2) *This paper would be the third publication about SAIGE, after the original single-point method and the subsequent SAIGE-GENE paper. Yet the software feels like it is still very much in beta stage. The documentation is clunky, the wiki has dead links, and the package content is littered with debug code. Some data objects are pasted to stdout, there are messages such as "DEBUG1" "DEBUG2" "DEBUG3", etc. Options naming is inconsistent particularly for binary options and flags (--IsBinaryOption TRUE and --BinaryOption TRUE exist). If this software is to be considered as mature, it needs to look like one, with consistent options naming, a robust and detailed documentation, a log argument and a verbosity argument. There hasn't been a release on github since version 0.29.5, please add subsequent version releases up until 0.44.6.5. Provide documentation and a suitable user-friendly interface to the CCT function(perhaps rename it also).*

RE: We appreciate the reviewer's suggestion. We have now completed a stable version (v1.0) of the program with the documentation provided <https://saigegit.github.io/SAIGE-doc/> with properly removing DEBUG outputs. We converted R code to C++ code for the score tests, SPA tests, and the CCT function, so the overhead between switching coding platforms was reduced, while the R interface remained for R package users. We incorporated the CCT function in the package, so users can obtain the Cauchy combination results by default when multiple variant sets were used to test.

We have removed several unnecessary options. For example, the binary option --IsSparse, which was primarily used for evaluating the program performance by developers, was removed, instead the sparsity of the genotype vector for less frequent variants was by default exploited to speed up the

program. The binary option `-IsOutputPvalueNAinGroupTestforBinary` for whether to output p-values without accounting for case-control imbalance has been removed and those p-values are output by default. `-IsOutputBETASEinBurdenTest` for whether to output the effect sizes and standard errors for Burden tests was removed and those estimates were output by default. For single-variant association tests, multiple options for output (`--IsOutputAFinCaseCtrl` and `-IsOutputNinCaseCtrl`) allele frequencies and sample sizes in cases and controls for binary traits were removed, and those values are output by default. Also, we have added options `--maxMAF_in_groupTest`, `--maxMAC_in_groupTest`, and `--annotation_in_groupTest` to allow specifying multiple MAF and MAC cutoffs and functional annotations in a single job. The option `-numLinesOutput` was replaced by `--markers_per_chunk` and `--groups_per_chunk` to specify the number of markers and sets to test and output as one chunk in the single-variant and set-based tests, respectively.

We have created a new program github page <https://github.com/saigegit/SAIGE> with the v1.0.0 release. The stable version of the program will be maintained by multiple SAIGE developers.

3) There are currently 149 open issues on the github page, many of which have not been answered. There is obviously a lot of interest in this software, yet the authors seem to struggle to provide support. The authors should solve and close as many issues as possible before releasing a published update. I also note that many closed issues did not result in code changes such as, for example, to provide more informative error messages.

RE: We thank the reviewer for pointing it out and sorry for not timely supporting the users. As is mentioned, the program evolves fast per many requests for new features and options, it is challenging to keep tracking issues for different versions. We have replied and closed many of the issues. We now create a new program github page and have sent messages to all users who left open issues to check the new website and welcome their comments and issue reports. In addition, we have noticed the challenge in installing the third party libraries that our program depends on for reading files. Version 1.0.0 no longer depends on the bgen library for reading bgen files.

4) One of the main advantage of SAIGE-GENE compared to its more performant competitors like REGENIE2 is that SAIGE-GENE allows to perform optimal tests (SKAT-O) and allows functional weights. Yet the paper does not explain how functional variant weights are collapsed when ultra-rare variants are collapsed as part of this scheme. Please add a paragraph detailing how this is done.

RE: We apologize for the unclear description of how the functional variant weights were incorporated when ultra-rare variants were collapsed. We have expanded the paragraph to detail how it is done in the “Collapsing ultra-rare variants” subsection in the Methods section and Supplementary Figure 15 to demonstrate the process.

“Ultra-rare variants with $MAC \leq 10$ were collapsed to a single marker (**Supplementary Figure 15**). More specifically, if no per-marker weights are provided by the user, all ultra-rare variants will be collapsed to a new variant in the “absence and presence” way¹⁰. The dosage for each sample was assigned as the maximum dosage value among all ultra-rare variants carried by the sample, if any. Then the weights of the collapsed variant and non ultra-rare variants ($MAC > 10$) are calculated based on their MAF from beta distribution $Beta(MAF, a_1, a_2)$. By default, same as the setting of the SKAT-O test, $a_1 = 1$ and $a_2 = 25$ are used. If the per-marker weights are provided by the user, the ultra-rare variants will be collapsed to a new variant whose dosages are the maximum values among the weighted dosages.”

Minor Comments

1) line 46-47, add here that SAIGE-GENE performs SKAT-(O) tests which are agnostic to genetic architecture. It is also the only one performing weighted tests at scale. Mention and compare the big competition of SAIGE-GENE, which is REGENIE2.

RE: Again, we thank the reviewer for pointing this out. We have added more information about the comparison with REGENIE2 in the Supplementary Note D.

“We observed that the Burden test p-values by SAIGE-GENE+ are highly concordant with the p-values by REGENIE2 (Pearson’s correlation $R^2 = 0.99$ for $-\log_{10}(p\text{-value})$) (Supplementary Figure 9). We also compared the empirical computation cost of SAIGE-GENE+ and REGENIE2². In Step 1 for fitting null models, SAIGE-GENE+ with a full GRM was more efficient than REGENIE2 (Supplementary Figure 7A and Supplementary Table 2). Out of the five runs with 150,000 samples that were randomly sub-sampled from the UK Biobank WES data with White British participants for glaucoma (1,741 cases and 162,408 controls) from the UKBB, the median computation time for Step 1 is 11 CPU hours using SAIGE-GENE+ and 36.5 CPU hours using REGENIE2 and the median memory usage is 5.4 Gb in SAIGE-GENE+ and 7.3 Gb in REGENIE2. Moreover, when a sparse GRM instead of a full GRM is used in Step 1 in SAIGE-GENE+, the time cost and memory usage dramatically dropped (< 1 min and 0.61Gb) . In Step 2, similar computation cost was observed for the two methods for Burden tests (Supplementary Figure 7B and Supplementary Table 3): 8.8 CPU hours and 0.93Gb by REGENIE2 and 9.1 CPU hours and 0.97Gb by SAIGE-GENE+, which additionally output the p-values by the Cauchy combination. SAIGE-GENE+ conducts the SKAT-O test, while REGENIE2 conducts Burden tests only and does not allow for incorporating marker level weights. Although the SKAT-O tests in SAIGE-GENE+ required nearly 6-7x more computation time (60 CPU hours) and 2x more memory (2.0 Gb) (Supplementary Figure 7B and Supplementary Table 3), through simulation studies, we observed that SKAT-O tests have higher power than Burden tests in all different scenario (Supplementary Table 6) with more significant p-values (Supplementary Figure 2) and higher median Chi-square statistics (Supplementary Table 7). ”

When we completed this revision, REGENIE3 was just released. As in response to the editor point 5, we did not formally include REGENIE3 comparison, as it basically implemented our approach, and the code may not be fully tested as shown in the inflated QQ plots.

2) l. 115. remove "the" age at menopause.

Done.

3) l. 116 remove "the" lymphocyte count.

Done.

4) l. 126 do not use "insignificant". Use "non-significant" or "miss the p-value threshold" or similar.

17Done.

5) I.132 conducted "on" LoF variants, not "to".

Done

Reviewer #3:

Remarks to the Author:

The authors present an improvement upon SAIGE-GENE that will be useful to the community. The evidence showing faster computational times and reduction of test statistic inflation when variants are very rare is really helpful.

1. However, I felt the paper was lacking details in why and how SAIGE-GENE+ performed better than SAIGE-GENE, as in what changed about the methods to create such an improvement. I noticed much focus on the 'ultra-rare' variant analysis with $MAC < 10$, but that doesn't explain why things performed better for $MAC < 0.1\%$ cutoff as well.

RE: We very much appreciate the reviewer's comment. The example we included to demonstrate that testing variant sets with lower MAF cutoffs may help identify novel associations is the association between the gene *PDCD1LG* and chronic lymphoid leukemia. The association is more significant with a lower MAF cutoff than $MAF \leq 1\%$. The marker-level associations and the associations with each MAF cutoff and functional annotation mask can be found at the browser <https://ukb-200kexome.leelabsg.org/assoc/PDCD1LG2/204.12>

The underlying reason for this observation could be that the associations are largely enriched in the rarer variants in the gene, e.g. the most significant variant is the low frequency variant rs7854303 (9:5557672, $MAF = 3.4 \times 10^{-4}$). Using a higher MAC cutoff 1% will include many non-causal variants,

18which decreases the power of the set-based association tests. Given that the underlying genetic architecture is usually unknown for different genes and phenotypes, incorporating multiple MAF cutoffs and functional annotations would lead to higher power to identify associations than using a single MAF cutoff or functional annotation. We have also conducted power simulation studies to illustrate it. The plots showing p-value comparisons are presented in Supplementary Figure 10 and the medians of the Chi-square test statistics are included in Supplementary Table 7.

2. I also didn't understand why the p-value improved for BRCA from $\times 10^{-3}$ to $\times 10^{-8}$ when the same MAF cutoff was used; same for GCK, as here: "For example, BRCA2 for breast cancer with MAF 0.1% has p-value 7.62×10^{-8} in SAIGE-GENE+ and 1.65×10^{-3} 96 in SAIGE-GENE, and GCK for diabetes with MAF 0.1% p-value 1.22×10^{-13} in SAIGE-GENE+ and 4.06×10^{-6} 97 in SAIGE-GENE." Why would the p-value improve if the MAF cutoff was the same?

RE: Again, we appreciate the reviewer's comment. For the association between BRCA and breast cancer, it can be seen at the browser (https://ukb-200kexome.leelabsg.org/assoc/BRCA2/20001_1002) that the associations are highly enriched in the ultra-rare LoF variants that tend to have the same effect directions. The single variant that was generated by collapsing all 142 ultra-rare LoF variants has p-value 4.9×10^{-22} .

Given the assumptions of the Burden and SKAT tests, it is known that the Burden test is more powerful than the SKAT test when most of the genetic variants in the test set are causal (having non-zero effects) with the same effect direction, whereas the SKAT test is more powerful when a small proportion of genetic variants are causal with inconsistent effect directions. Without collapsing, the Burden test (p-value = 0.000738) has a more significant p-value than the SKAT test (p-value = 0.114). But because only a proportion of variants are causal, Burden test can still suffer from the low association power. After collapsing, the SKAT test (p-value = 2.671×10^{-8}) has a more significant p-value than the Burden test (p-value = 0.00452). This could be because the association signal is largely contributed by the collapsed ultra-rare variants and the SKAT test is more robust to the small proportion of causal variants in the test sets. Similarly, it is observed that the association between the gene GCK and diabetes is driven by the collapsed ultra-rare variants <https://ukb-200kexome.leelabsg.org/assoc/GCK/2443>.

We have added the Supplementary Note E “*BRCA2* for breast cancer with MAF \leq 0.1% had p-value 7.62×10^{-8} in SAIGE-GENE+ and 1.65×10^{-3} in SAIGE-GENE. Similarly, we observed the gene *GCK* for diabetes with maximum MAF 0.1% had a more significant p-value (1.22×10^{-13}) in SAIGE-GENE+ than in SAIGE-GENE (p-value = 4.06×10^{-6}).

In *BRCA2*-Breast Cancer, the associations are highly enriched in the ultra-rare LoF variants that tend to have the same effect directions, as is observed that the collapsed variant from 142 ultra-rare LoF variants had p-value 4.9×10^{-22} (https://ukb-200kexome.leelabsg.org/assoc/BRCA2/20001_1002). It is known that the Burden test is more powerful than the SKAT³ test when most of the genetic variants in the test set are causal (having non-zero effects) with the same effect direction, whereas the SKAT test is more powerful when a small proportion of genetic variants are causal with inconsistent effect directions. Without collapsing in SAIGE-GENE, the Burden test p-value (0.000738) is more significant than the SKAT test p-value (0.114). But because only a small proportion of variants are causal, Burden test can still suffer from the low association power. After collapsing in SAIGE-GENE+, the SKAT test (p-value= 2.67×10^{-8}) had a more significant p-value than the Burden test (p-value = 0.00452). This could be because the association signal is largely contributed by the collapsed ultra-rare variants and the SKAT test is more robust to the large proportion of non-causal variants in the test sets. Similarly, it is observed that the association between the gene *GCK* and diabetes is driven by the collapsed ultra-rare variants <https://ukb-200kexome.leelabsg.org/assoc/GCK/2443>.

”
-

3. There was also a glaring omission of regenie in this manuscript. regenie, which was based on SAIGE, has really arisen as the key gene-based collapsing analysis method this past year, and a comparison to regenie would really show how SAIGE-GENE+ is an improvement.

RE: We thank the reviewer for the insightful suggestion. REGENIE2 implemented the Burden test, which collapses all genetic variants in the test set, assuming that most variants are causal and effect sizes are consistent. While the Burden test can be more powerful when the assumption is valid, it can

20have much lower power than non-burden type tests like SKAT when the assumption is not satisfied. SKAT-O combines both Burden and SKAT to achieve robust power. In SAIGE-GENE and SAIGE-GENE+, SKAT-O is performed by default with an option provided to only conduct the Burden test. For each testing gene or region, multiple function annotations and maximum MAF cutoffs can be used, followed by using a Cauchy combination to calculate a combined p-value.

Through extensive simulation studies, we showed that 1. SAIGE-GENE+ provides consistent association p-values of Burden tests as REGENIE2 (Pearson's correlation $R^2 = 0.99$ for $-\log_{10}(p\text{-value})$) (Supplementary Figure 9). 2. The SKAT-O tests in SAIGE-GENE+ have higher power than Burden tests with more significant p-values and higher median Chi-square statistics (Supplementary Figure 2, Supplementary Table 7).

We have also evaluated the computation cost of running SAIGE-GENE+ and REGENIE for set-based tests using the UKBB WES data. Both SAIGE-GENE+ and REGENIE2 require two steps, Step 1 for fitting the null model and step2 for testing associations for each set, so we measure the computation cost separately. In step1, we have found that our software achieves shorter computation time and lower memory usages than REGENIE2 (Supplementary Figure 7). The computation cost gap was much larger when the sparse GRM was used in SAIGE-GENE+ (SAIGE GENE+: <1 min, and 0.61Gb vs REGENIE2: 36.5 CPU hours and 7.3 Gb). In Step 2, similar computation cost was observed for the two programs when conducting Burden tests only (Supplementary Figure 7B and Supplementary Table 3): 8.8 CPU hours and 0.93Gb by REGENIE2 and 9.1 CPU hours and 0.97Gb by SAIGE-GENE+.

We added following texts in the manuscript to present the result.

In the Main section:

“Our results also demonstrated as previously reported that the SKAT-O test can have higher power than the Burden test with more significant p-values and higher median Chi-square statistics (Supplementary Figure 2, Supplementary Table 7), while the Burden test p-values from SAIGE-GENE+ are highly concordant to p-values from REGENIE2 (Pearson's correlation $R^2 = 0.99$ for $-\log_{10}(p\text{-value})$) (Supplementary Figure 9).”

In the Supplementary Note D:

"We observed that the Burden test p-values by SAIGE-GENE+ are highly concordant with the p-values by REGENIE2 (Pearson's correlation $R^2 = 0.99$ for $-\log_{10}(p\text{-value})$) (Supplementary Figure 9). We also compared the empirical computation cost of SAIGE-GENE+ and REGENIE2². In Step 1 for fitting null models, SAIGE-GENE+ with a full GRM was more efficient than REGENIE2 (Supplementary Figure 7A and Supplementary Table 2). Out of the five runs with 150,000 samples that were randomly sub-sampled from the UK Biobank WES data with White British participants for glaucoma (1,741 cases and 162,408 controls) from the UKBB, the median computation time for Step 1 is 11 CPU hours using SAIGE-GENE+ and 36.5 CPU hours using REGENIE2 and the median memory usage is 5.4 Gb in SAIGE-GENE+ and 7.3 Gb in REGENIE2. Moreover, when a sparse GRM instead of a full GRM is used in Step 1 in SAIGE-GENE+, the time cost and memory usage dramatically dropped (< 1 min and 0.61Gb). In Step 2, similar computation cost was observed for the two methods for Burden tests (Supplementary Figure 7B and Supplementary Table 3): 8.8 CPU hours and 0.93Gb by REGENIE2 and 9.1 CPU hours and 0.97Gb by SAIGE-GENE+, which additionally output the p-values by the Cauchy combination. SAIGE-GENE+ conducts the SKAT-O test, while REGENIE2 conducts Burden tests only and does not allow for incorporating marker level weights. Although the SKAT-O tests in SAIGE-GENE+ required nearly 6-7x more computation time (60 CPU hours) and 2x more memory (2.0 Gb) (Supplementary Figure 7B and Supplementary Table 3), through simulation studies, we observed that SKAT-O tests have higher power than Burden tests in all different scenario (Supplementary Table 6) with more significant p-values (Supplementary Figure 2) and higher median Chi-square statistics (Supplementary Table 7)."

When we completed this revision, REGENIE3 was just released. As in response to the editor point 5, we did not formally include REGENIE3 comparison, as it basically implemented our approach, and the code may not be fully tested as shown in the inflated QQ plots.

Decision Letter, first revision:

17th June 2022

Dear Seunggeun,

Your revised manuscript "Set-based rare variant association tests for biobank scale sequencing data sets" (NG-BC57699R1) has been seen by two of the original referees. As you will see from their comments below, they find that the paper has improved in revision, and therefore we will be happy in principle to publish it in Nature Genetics as a Brief Communication pending final revisions to satisfy

22Reviewer #2's remaining concerns and to comply with our editorial and formatting guidelines.

We are now performing detailed checks on your paper, and we will send you a checklist detailing our editorial and formatting requirements soon. Please do not upload the final materials and make any revisions until you receive this additional information from us.

Thank you again for your interest in Nature Genetics. Please do not hesitate to contact me if you have any questions.

Sincerely,
Kyle

Kyle Vogan, PhD
Senior Editor
Nature Genetics
<https://orcid.org/0000-0001-9565-9665>

Reviewer #2 (Remarks to the Author):

In this revision, the authors have addressed the limited set of comments that have been highlighted by the editor. The comparison with REGENIE2, as well as an improved documentation and code, are welcome changes. I remain unconvinced that the scale of the work presented here, which corresponds to a modest update to existing software, warrants re-publication under the current title.

Before publication, I would like to point out that the manuscript requires thorough proofreading. There are many grammatical mishaps and clumsy sentences that broadly hamper the clarity of the paper. There are too many to point all of them out here, however here are some examples:

I.25 "the" large-scale

I.35 write "recently released whole-exome sequencing (WES) data for most of their participants"

I.36 "However, best practices remain"

I.44 write "minimum p-value or Cauchy combination methods."

I.46 write "conduct variance-component set-based rare variant tests"

I.49 Please give more background and explanation, and rewrite for clarity: "Burden tests (such as implemented by Regenie 2), collapse multiple rare variant genotypes to a single one, allowing the use of single-variant tests. However, those can suffer from power loss compared to SKAT-O tests, which are more robust to different proportions of causal variants and heterogeneity of causal effect directions. This was confirmed in our simulation studies."

I.53 The sentence starting here with "SAIGE-GENE" feels out of place and it is not clear why the sample size drops from 281,850 here to 160K in the next. Please make an effort to keep your train of thought clear to the reader.

Please apply the same proofreading approach to the rest of the manuscript.

23This new method essentially collapses ultra-rare variants in a burden test to reduce computational burden, then performs an optimal/unified test including this synthetic genotype and actual genotypes for less rare variants. This is not mentioned as clearly anywhere in the paper. It should be, for example, around lines 73-76, when describing the "absence-and-presence" approach, as well as in the first paragraph of the methods.

This first paragraph of the methods is surprisingly unclear. As far as I understand, in the absence of weights, the authors use a simple absence-or-presence burden for collapsing of ultra-rare variants. When dosages are present instead of hard called genotypes, the maximum across all ultra-rare variant dosages is used. When weights are present, the weights are multiplied by the genotype or dosage and the maximum is used as a weighted dosage for that sample. Is this correct? If yes, the authors should probably write something along those lines.

Reviewer #3 (Remarks to the Author):

The authors have addressed my comments.

Our ref: NG-BC57699R1

29th June 2022

Dear Seunggeun,

Thank you for your patience as we've prepared the guidelines for final submission of your Nature Genetics manuscript "Set-based rare variant association tests for biobank scale sequencing data sets" (NG-BC57699R1). Please carefully follow the step-by-step instructions provided in the attached file and add a response in each row of the table to indicate the changes that you have made. Ensuring that each point is addressed will help to ensure that your revised manuscript can be swiftly handed over to our production team.

In recognition of the time and expertise our reviewers provide to our editorial process, we would like to formally acknowledge their contribution to the external peer review of your manuscript entitled "Set-based rare variant association tests for biobank scale sequencing data sets". For those reviewers

24who give their assent, we will be publishing their names alongside the published article.

Nature Genetics offers a Transparent Peer Review option for new original research manuscripts submitted after December 1st, 2020. As part of this initiative, we encourage our authors to support increased transparency into the peer review process by agreeing to have the reviewer comments, author rebuttal letters, and editorial decision letters published as a Supplementary item. When you submit your final files please clearly state in your cover letter whether or not you would like to participate in this initiative. Please note that failure to state your preference will result in delays in accepting your manuscript for publication.

Cover suggestions

As you prepare your final files, we encourage you to consider whether you have any images or illustrations that may be appropriate for use on the cover of Nature Genetics.

We accept TIFF, JPEG, PNG or PSD file formats (a layered PSD file would be ideal), and the image should be at least 300 ppi resolution (preferably 600-1200 ppi), in CMYK color mode.

Nature Genetics has now transitioned to a unified Rights Collection system which will allow our Author Services team to quickly and easily collect the rights and permissions required to publish your work. Approximately 10 days after your paper is formally accepted, you will receive an email in providing you with a link to complete the grant of rights. If your paper is eligible for Open Access, our Author Services team will also be in touch regarding any additional information that may be required to arrange payment for your article.

Please note that Nature Genetics is a Transformative Journal (TJ). Authors may publish their research with us through the traditional subscription access route or make their paper immediately open access through payment of an article-processing charge (APC). Authors will not be required to make a final decision about access to their article until it has been accepted. [Find out more about Transformative Journals](https://www.springernature.com/gp/open-research/transformative-journals)

Authors may need to take specific actions to achieve [compliance](https://www.springernature.com/gp/open-research/funding/policy-compliance-faqs) with funder and institutional open access mandates. If your research is supported by a funder that requires immediate open access (e.g. according to [Plan S principles](https://www.springernature.com/gp/open-research/plan-s-compliance)) then you should select the gold OA route, and we will direct you to the compliant route where

25possible. For authors selecting the subscription publication route, the journal's standard licensing terms will need to be accepted, including [self-archiving policies](https://www.springernature.com/gp/open-research/policies/journal-policies). Those licensing terms will supersede any other terms that the author or any third party may assert apply to any version of the manuscript.

Please use the following link to upload your final submission files:

[REDACTED]

Best wishes,
Kyle

Kyle Vogan, PhD
Senior Editor
Nature Genetics
<https://orcid.org/0000-0001-9565-9665>

Reviewer #1:
None

Reviewer #2:
Remarks to the Author:

In this revision, the authors have addressed the limited set of comments that have been highlighted by the editor. The comparison with REGENIE2, as well as an improved documentation and code, are welcome changes. I remain unconvinced that the scale of the work presented here, which corresponds to a modest update to existing software, warrants re-publication under the current title.

Before publication, I would like to point out that the manuscript requires thorough proofreading. There are many grammatical mishaps and clumsy sentences that broadly hamper the clarity of the paper. There are too many to point all of them out here, however here are some examples:

l.25 "the" large-scale

l.35 write "recently released whole-exome sequencing (WES) data for most of their participants"

26I.36 "However, best practices remain"

I.44 write "minimum p-value or Cauchy combination methods."

I.46 write "conduct variance-component set-based rare variant tests"

I.49 Please give more background and explanation, and rewrite for clarity: "Burden tests (such as implemented by Regenie 2), collapse multiple rare variant genotypes to a single one, allowing the use of single-variant tests. However, those can suffer from power loss compared to SKAT-O tests, which are more robust to different proportions of causal variants and heterogeneity of causal effect directions. This was confirmed in our simulation studies."

I.53 The sentence starting here with "SAIGE-GENE" feels out of place and it is not clear why the sample size drops from 281,850 here to 160K in the next. Please make an effort to keep your train of thought clear to the reader.

Please apply the same proofreading approach to the rest of the manuscript.

This new method essentially collapses ultra-rare variants in a burden test to reduce computational burden, then performs an optimal/unified test including this synthetic genotype and actual genotypes for less rare variants. This is not mentioned as clearly anywhere in the paper. It should be, for example, around lines 73-76, when describing the "absence-and-presence" approach, as well as in the first paragraph of the methods.

This first paragraph of the methods is surprisingly unclear. As far as I understand, in the absence of weights, the authors use a simple absence-or-presence burden for collapsing of ultra-rare variants. When dosages are present instead of hard called genotypes, the maximum across all ultra-rare variant dosages is used. When weights are present, the weights are multiplied by the genotype or dosage and the maximum is used as a weighted dosage for that sample. Is this correct? If yes, the authors should probably write something along those lines.

Author Rebuttal, first revision:

Response to Reviewers Comments

Title: Set-based rare variant association tests for biobank scale sequencing data sets

We thank the reviewers and editors for their thoughtful and constructive comments that helped greatly improve the manuscript. Below are our detailed responses (in bold). Quoted text from the manuscript is highlighted in blue. To distinguish comments and responses, comments are italicized. In the main manuscript and supplementary materials, added or revised text is highlighted in yellow.

27Reviewer #2 (Remarks to the Author):

In this revision, the authors have addressed the limited set of comments that have been highlighted by the editor. The comparison with REGENIE2, as well as an improved documentation and code, are welcome changes. I remain unconvinced that the scale of the work presented here, which corresponds to a modest update to existing software, warrants re-publication under the current title.

Before publication, I would like to point out that the manuscript requires thorough proofreading. There are many grammatical mishaps and clumsy sentences that broadly hamper the clarity of the paper.

There are too many to point all of them out here, however here are some examples:

RE: We thank the reviewer for this important suggestion. We have done proofreading to the entire manuscript and fixed grammatical mishaps and clumsy sentences, including examples listed below.

l.25 "the" large-scale

RE: We removed "the" from the sentence (line 37) and now it reads "UK Biobank has released large-scale whole-exome sequencing (WES) data"

l.35 write "recently released whole-exome sequencing (WES) data for most of their participants"

RE: We removed "the" from the sentence (line 47) and now it reads "UK Biobank (UKBB) recently released whole-exome sequencing (WES) data¹, allowing for studying rare variant associations for complex phenotypes"

l.36 "However, best practices remain"

RE: We removed “the” from the sentence (line 47 - 48) and now it reads “However, best practices remain unclear for rare variant tests in large-scale biobanks.”

I.44 write “minimum p-value or Cauchy combination methods.”

RE: We removed “the” from the sentence (line 53 - 55) and now it reads “To incorporate multiple MAF cutoffs and functional annotations, multiple tests are needed for each gene or region, and results need to be combined using minimum p-value or Cauchy combination method^{2,3}”

I.46 write “conduct variance-component set-based rare variant tests”

RE: Now it reads (lines 57 - 58) “Currently, SAIGE-GENE⁴ is the only method developed to conduct variance-component set-based tests, such as SKAT⁵ and SKAT-O⁶, for unbalanced case-control phenotypes in biobank-scale data.”

I.49 Please give more background and explanation, and rewrite for clarity: “Burden tests (such as implemented by Regenie 2), collapse multiple rare variant genotypes to a single one, allowing the use of single-variant tests. However, those can suffer from power loss compared to SKAT-O tests, which are more robust to different proportions of causal variants and heterogeneity of causal effect directions. This was confirmed in our simulation studies.”

RE: Now it reads (lines 60 - 63) “Burden tests (such as implemented in REGENIE⁷) collapse multiple rare variants into a single variant, allowing the use of well-developed single-variant tests. However, Burden tests can have low power compared to SKAT and SKAT-O⁶. This was confirmed in our simulation studies (Extended Data Fig. 2).”

I.53 The sentence starting here with “SAIGE-GENE” feels out of place and it is not clear why the sample size drops from 281,850 here to 160K in the next. Please make an effort to keep your train of thought clear to the reader.

RE: Now it reads (lines 63 - 67) “In analyses of UKBB WES data of 160K white British individuals (from the release with 200k individuals), we found that SAIGE-GENE performed well when testing variants with $MAF \leq 1\%$ (Figure 1A), but inflation was observed in SKAT and SKAT-O in SAIGE-GENE when restricting to variants with $MAF \leq 0.1\%$ or 0.01% and the case-control ratios were more unbalanced than 1:30 (Figure 1A, Extended Data Fig. 3).”

Please apply the same proofreading approach to the rest of the manuscript.

RE: We have done proofreading to the entire manuscript

This new method essentially collapses ultra-rare variants in a burden test to reduce computational burden, then performs an optimal/unified test including this synthetic genotype and actual genotypes for less rare variants. This is not mentioned as clearly anywhere in the paper. It should be, for example, around lines 73-76, when describing the "absence-and-presence" approach, as well as in the first paragraph of the methods.

RE: Thanks for pointing this out. Because the approach used in SAIGE-GENE+ to collapse ultra-rare variants is slightly different from the "absence-and-presence" approach, we removed it from the text. In lines 79 - 82 “To reduce data sparsity due to ultra-rare variants, prior to testing each variant set, SAIGE-GENE+ collapses variants with $MAC \leq 10$ and then tests the collapsed variant together with all other variants with $MAC \geq 10$ (Extended Data Fig. 5, Methods). Collapsing has been commonly used for ultra-rare variants^{8,9} by assuming those variants have the same direction of effects on phenotypes.”

This first paragraph of the methods is surprisingly unclear. As far as I understand, in the absence of weights, the authors use a simple absence-or-presence burden for collapsing of ultra-rare variants. When dosages are present instead of hard called genotypes, the maximum across all ultra-rare variant dosages is used. When weights are present, the weights are multiplied by the genotype or dosage and the maximum is used as a weighted dosage for that sample. Is this correct? If yes, the authors should probably write something along those lines.

RE: We apologize for the unclear description and have rewritten the Collapsing ultra-rare variants subsection of the Methods section.

“Ultra-rare variants with $MAC \leq 10$ are collapsed to a single marker, as illustrated in **Extended Data Fig. 5**. Same as SAIGE-GENE, SAIGE-GENE+ allows incorporating weights for dosages or hard called genotypes of each marker. By default, to up-weight rare variants, SAIGE-GENE+ calculates the weight for each variant using its MAF from a beta distribution $Beta(MAF, 1, 25)$. SAIGE-GENE+ also allows users to specify per-marker weights. The weighting scheme when collapsing ultra-rare variants is slightly different between these two. 1. If the default MAF-based beta-weight is used, SAIGE-GENE+ first obtains the collapsed variant and assigns the weight based on collapsed variant frequency (**Extended Data Fig. 5A**). In particular, the dosage or genotype for each sample of the collapsed variant is assigned as the maximum raw dosage or genotype value among all ultra-rare variants carried by the sample. Then the weights of the collapsed variant and other less rare variants ($MAC > 10$) are calculated based on their MAF. 2. If the per-marker weights are provided by users (**Extended Data Fig. 5B**), the dosages or genotypes of the ultra-rare variants are first multiplied by the provided weights and then collapsed to a new variant whose dosage or genotype for each sample is assigned as the maximum values among the weighted dosages or genotypes of all ultra-rare variants carried by the sample. SAIGE-GENE+ also allows not incorporating any weights to set-based tests and collapses ultra-rare variants following the second scheme described above as it is a special case which has equal weights for all variants.”

Extended Data Fig. 5 is used to illustrate the two scenarios.

- A. No per-marker weights are provided in the group file by the user. The weights of the collapsed variant and other non-ultra-rare variants ($MAC > 10$) are calculated based on their MAFs from Beta distribution $w_j = \text{Beta}(MAF_j, a_1, a_2)$. By default, $a_1 = 1, a_2 = 25$.

	Ultra-rare variants (MAC ≤ 10)	Collapsed (max of dosages)	
Sample 1	0 0 0 0 0 0 0 0 0 0	→ 0	Weight of the collapsed variant $w = \text{Beta}(MAF, a_1, a_2)$
Sample 2	0 1 0 0 0 0 0 0 0 0	→ 1	
Sample 3	1 0 0 0 0 0 0 0 1 0	→ 1	
Sample 4	0 0 0 2 0 0 0 0 1 0	→ 2	
Sample 5	0 0 0 0 1 0 1 0 0 0	→ 1	

- B. The per-marker weights are provided in the group file by the user

	Ultra-rare variants (MAC ≤ 10)	Collapsed (max of the weighted dosages)
Sample 1	0 0 0 0 0 0 0 0 0 0	→ 0
Sample 2	0 1 0 0 0 0 0 0 0 0	→ 0.2
Sample 3	1 0 0 0 0 0 0 0 1 0	→ $\max(0.1, 0.9) = 0.9$
Sample 4	0 0 0 2 0 0 0 0 1 0	→ $\max(2 \cdot 0.4, 0.9) = 0.9$
Sample 5	0 0 0 0 1 0 1 0 0 0	→ $\max(0.5, 0.7) = 0.7$
User-specified weight	0.1 0.2 0.3 0.4 0.5 0.6 0.7 0.8 0.9 1	

Final Decision Letter:

Subject: Decision on NG-BC57699R2 Lee

In reply please quote: NG-BC57699R2 Lee

29th July 2022

Dear Seunggeun,

I am delighted to say that your manuscript "SAIGE-GENE+ improves the efficiency and accuracy of set-based rare variant association tests" has been accepted for publication in an upcoming issue of Nature Genetics.

Over the next few weeks, your paper will be copyedited to ensure that it conforms to Nature Genetics style. Once your paper is typeset, you will receive an email with a link to choose the appropriate

32publishing options for your paper and our Author Services team will be in touch regarding any additional information that may be required.

Your paper will be published online after we receive your corrections and will appear in print in the next available issue. You can find out your date of online publication by contacting the Nature Press Office (press@nature.com) after sending your e-proof corrections. Now is the time to inform your Public Relations or Press Office about your paper, as they might be interested in promoting its publication. This will allow them time to prepare an accurate and satisfactory press release. Include your manuscript tracking number (NG-BC57699R2) and the name of the journal, which they will need when they contact our Press Office.

Before your paper is published online, we will be distributing a press release to news organizations worldwide, which may very well include details of your work. We are happy for your institution or funding agency to prepare its own press release, but it must mention the embargo date and Nature Genetics. Our Press Office may contact you closer to the time of publication, but if you or your Press Office have any enquiries in the meantime, please contact press@nature.com.

Please note that Nature Genetics is a Transformative Journal (TJ). Authors may publish their research with us through the traditional subscription access route or make their paper immediately open access through payment of an article-processing charge (APC). Authors will not be required to make a final decision about access to their article until it has been accepted. [Find out more about Transformative Journals](https://www.springernature.com/gp/open-research/transformative-journals)

Authors may need to take specific actions to achieve [compliance with funder and institutional open access mandates](https://www.springernature.com/gp/open-research/funding/policy-compliance-faqs). If your research is supported by a funder that requires immediate open access (e.g. according to [Plan S principles](https://www.springernature.com/gp/open-research/plan-s-compliance)),

33then you should select the gold OA route, and we will direct you to the compliant route where possible. For authors selecting the subscription publication route, the journal's standard licensing terms will need to be accepted, including <https://www.nature.com/nature-portfolio/editorial-policies/self-archiving-and-license-to-publish>. Those licensing terms will supersede any other terms that the author or any third party may assert apply to any version of the manuscript.

Please note that Nature Portfolio offers an immediate open access option only for papers that were first submitted after 1 January 2021.

If you have not already done so, we invite you to upload the step-by-step protocols used in this manuscript to the Protocols Exchange, part of our on-line web resource, natureprotocols.com. If you complete the upload by the time you receive your manuscript proofs, we can insert links in your article that lead directly to the protocol details. Your protocol will be made freely available upon publication of your paper. By participating in natureprotocols.com, you are enabling researchers to more readily reproduce or adapt the methodology you use. [Natureprotocols.com](http://natureprotocols.com) is fully searchable, providing your protocols and paper with increased utility and visibility. Please submit your protocol to <https://protocolexchange.researchsquare.com/>. After entering your nature.com username and password you will need to enter your manuscript number (NG-BC57699R2). Further information can be found at <https://www.nature.com/nature-portfolio/editorial-policies/reporting-standards#protocols>

Sincerely,

34Kyle

Kyle Vogan, PhD
Senior Editor
Nature Genetics
<https://orcid.org/0000-0001-9565-9665>

Click here if you would like to recommend Nature Genetics to your librarian
<http://www.nature.com/subscriptions/recommend.html#forms>

** Visit the Springer Nature Editorial and Publishing website at http://editorial-jobs.springernature.com?utm_source=ejp_NGen_email&utm_medium=ejp_NGen_email&utm_campaign=ejp_NGen for more information about our career opportunities. If you have any questions please click [here](mailto:editorial.publishing.jobs@springernature.com). **